# Hippocampal and Reticulo-Thalamic Parvalbumin Interneurons and Synaptic Re-Organization during Sleep Disorders in the Rat Models of Parkinson’s Disease Neuropathology

**DOI:** 10.3390/ijms22168922

**Published:** 2021-08-19

**Authors:** Ljiljana Radovanovic, Jelena Petrovic, Jasna Saponjic

**Affiliations:** Institute for Biological Research “Sinisa Stankovic”, National Institute of Republic of Serbia, Department of Neurobiology, University of Belgrade, 11060 Belgrade, Serbia; ljiljana.radovanovic@ibiss.bg.ac.rs

**Keywords:** Parkinson’s disease, rat, prodromal sleep disorders, sleep spindle dynamics, hippocampus, reticulo-thalamic nucleus, parvalbumin-expressing interneurons, microtubule-associated protein 2 (MAP2), postsynaptic density protein 95 (PSD-95)

## Abstract

We investigated the alterations of hippocampal and reticulo-thalamic (RT) GABAergic parvalbumin (PV) interneurons and their synaptic re-organizations underlying the prodromal local sleep disorders in the distinct rat models of Parkinson’s disease (PD). We demonstrated for the first time that REM sleep is a predisposing state for the high-voltage sleep spindles (HVS) induction in all experimental models of PD, particularly during hippocampal REM sleep in the hemiparkinsonian models. There were the opposite underlying alterations of the hippocampal and RT GABAergic PV+ interneurons along with the distinct MAP2 and PSD-95 expressions. Whereas the PD cholinopathy enhanced the number of PV+ interneurons and suppressed the MAP2/PSD-95 expression, the hemiparkinsonism with PD cholinopathy reduced the number of PV+ interneurons and enhanced the MAP2/PSD-95 expression in the hippocampus. Whereas the PD cholinopathy did not alter PV+ interneurons but partially enhanced MAP2 and suppressed PSD-95 expression remotely in the RT, the hemiparkinsonism with PD cholinopathy reduced the PV+ interneurons, enhanced MAP2, and did not change PSD-95 expression remotely in the RT. Our study demonstrates for the first time an important regulatory role of the hippocampal and RT GABAergic PV+ interneurons and the synaptic protein dynamic alterations in the distinct rat models of PD neuropathology.

## 1. Introduction

Parkinson’s disease (PD) is the second most common neurodegenerative disease that predominantly affects the motor system as a result of dopaminergic (DA) neuronal loss within the substantia nigra pars compacta (SNpc) [1]. For decades, the clinical diagnosis of PD is dominantly based on identification of motor impairments such as bradykinesia, rigidity, resting tremor, and posture and gait problems [2]. Unfortunately, these motor symptoms appear rather late, following a substantial nigrostriatal dopaminergic neurodegeneration. It has been estimated that at least 50–80% of the nigrostriatal dopaminergic innervation is lost before the onset of clinical parkinsonism [2,3].

However, over the past few years, the concept of PD pathology has been changed: instead of being regarded as a motor disease induced by the selective loss of DA neurons within the SNpc, PD is now being recognized as a severe multisystem neurodegenerative disorder [4]. It has been suggested that apart from the dysfunction of dopaminergic system underlying the motor impairments, many other brain systems also undergo the pathological alterations and contribute to the non-motor symptoms [5,6]. Namely, the heterogeneous progression of PD neuropathology [7], with the clinical symptoms reflecting the localization and progression of underlying neuropathology [8], has been correlated to the gradual appearance of the non-motor dysfunctions, which are common, preclinical features of PD, linked to the initial (prodromal) stage of disease. Impaired olfaction, gastrointestinal abnormalities, sleep disturbances, visual alterations, and cognitive and mood disorders have been recognized as the non-motor manifestations that precede and accompany the classical motor impairments in PD [4,6].

Sleep disorders are the most frequent prodromal symptom of PD, with a prevalence of 60–70% [9]. Sleep-related symptoms of PD comprise a broad spectrum of conditions, which include insomnia, excessive daytime sleepiness [10], sleep fragmentation, restless leg syndrome [11], the difficulty in initiating and maintaining sleep, and the disturbances of rapid-eye movement (REM) sleep, such as REM sleep behavioral disorder (RBD) [9,12]. These sleep disorders progressively worsen in the course of PD and represent a major factor affecting the patient’s life [13]. Moreover, they are refractory to, or exacerbated by standard anti-parkinsonian medications, and they may promote the emergence of motor complications caused by a standard pharmacological therapy [9,14]. Still, only RBD, as a parasomnia characterized by the loss of normal muscle atonia during REM sleep and dream enactment behavior, has shown to be an early predictor of the development of PD [4,15].

In contrast to the well-defined pathophysiology of the motor impairments, the neuroanatomical and molecular substrates of PD non-motor symptoms, which are important in the early stages of PD, are far from clear. Current evidence suggests PD as a multisystem neurodegenerative syndrome where the neurotransmitters such as acetylcholine, serotonin, noradrenaline, glutamate, and gama-aminobutyric acid (GABA) play important pathophysiological roles [1,16]. Moreover, at the cellular level, PD presents synaptopathy [17].

Particularly, a dysregulation of the GABAergic system has received little attention, although the spectrum of non-motor symptoms might be linked to this pathway. GABA is the main inhibitory neurotransmitter in the central nervous system and a sleep-promoting neurotransmitter [18,19,20] that is primarily released by the local interneurons to regulate cortical and subcortical microcircuits. GABAergic signaling modulates a wide range of physiological functions, including sleep, sensory perception, information processing, and cognition [2]. There is evidence that the prodromal non-motor manifestations of PD are undoubtedly related to the downregulation of GABA neurotransmission [1,2]. Recently, the GABAergic dysregulation has been observed in the basal ganglia of patients with PD along with the striatal dopaminergic axons co-release of GABA [21]. In addition, it has been shown that GABA agonists relieve motor symptoms and protect dopaminergic cell bodies in the mice models of PD [22,23], as well as the RBD phenotype developed in the transgenic mice when the function of glycine and GABA receptors were impaired [24]. Although the downregulation of two key markers of GABAergic cells (glutamic acid decarboxylase-67 (GAD67) and calcium-binding protein parvalbumin (PV)) was evidenced in the dorsolateral prefrontal cortex of PD patients, without cell loss, the role of GABAergic neurotransmission in premotor stages of PD has not been established yet, which needs to be elucidated in future studies [1].

Our previous research suggests sleep disorders, particularly the REM sleep disorders, as the possible functional biomarkers of neurodegeneration that are relevant to PD and as the biomarkers of an earlier aging onset in the brain with neurodegeneration vs. physiological (healthy) brain. Namely, we previously evidenced in our rat model of PD cholinopathy (the bilateral lesion of the pedunculopontine tegmental nucleus (PPT); PPT lesion) the topographically differently expressed EEG microstructures within the sensorimotor and motor cortex during non-rapid eye movement (NREM) and REM sleep, alongside the appearance of two REM sleep states, particularly within the motor cortex. Distinct REM states were differential with regard to the EEG microstructures, the electromyographic (EMG) power, and the sensorimotor and motor cortical drives to the dorsal nuchal muscles [25,26]. These altered cortical drives were commonly expressed during both REM states as the impaired beta oscillation drive [27], but the sensorimotor cortical drive was altered more severely during “healthy” REM (REM with atonia, theta REM) than during the pathological REM sleep (REM without atonia, sigma REM). In addition, the hallmarks of an earlier aging onset during the PD cholinopathy were consistently expressed through the EEG sigma amplitude augmentation during REM sleep, as a unique and pathological REM sleep phenomenon, alongside the broadly altered motor cortical drive during NREM and REM sleep [28]. This pathological REM state has been shown to be the REM sleep “enriched” with sleep spindles, which is a unique phenomenon and a possible biomarker of earlier aging onset in the rat model of PD cholinopathy [29,30], and it suggests a disorder at the thalamo-cortical and hippocampal level. Namely, an impaired cholinergic innervation was expressed earlier as a sleep disorder than as a movement disorder; it was the earliest and long-lasting at the hippocampal and thalamo-cortical level, and it was followed by a delayed hypokinesia [30]. Overall, this study suggested that regarding how they occurred, the hippocampal NREM sleep disorder, an altered high voltage sleep spindle dynamics during REM sleep in the hippocampus and motor cortex, and hypokinesia may serve as the biomarkers of PD cholinopathy onset and progression [30]. In addition, our results in the animal model of PD cholinopathy are in accordance with the imaging studies in humans which demonstrated the thalamic cholinergic denervation in the Parkinsonian disorders with or without dementia and suggested that the neurodegenerative involvement of thalamic cholinergic afferent projections, arising from the PPT, may contribute to the disease specific motor and cognitive abnormalities [31].

Moreover, our previous studies in the rat models of hemiparkinsonism (the unilateral SNpc lesion; SNpc lesion; or combined unilateral SNpc lesion and bilateral PPT lesion; SNpc/PPT lesion) provided novel evidence for the importance of the SNpc dopaminergic innervation in sleep regulation, theta rhythm generation, and sleep spindle dynamics control, along with the importance of REM sleep regulatory substrate for sleep spindle generation and the cortico-hippocampal synchronizations of EEG oscillations [32]. Furthermore, in the rat models of hemiparkinsonism (SNpc lesioned rats and SNpc/PPT lesioned rats), we evidenced impaired spatial memory abilities followed by the severe hippocampal prodromal sleep disorders, which were expressed as the sleep fragmentation and distinct NREM/REM EEG microstructure alterations vs. the motor cortex; the opposite regulatory role of the dopaminergic vs. cholinergic control of the NREM delta and beta oscillation amplitudes in the hippocampus; and the important role of REM neurochemical substrate in the dopaminergic control of beta oscillations [33]. In addition, we recently demonstrated the brain structure-related and NREM/REM sleep-related heterogeneity of the simultaneous and non-simultaneous motor cortical and hippocampal local sleep in control rats [34], suggesting the importance of both the local neuronal network substrate and the NREM/REM neurochemical substrate in the control mechanisms of sleep in physiological as well as in any pathological condition [33,34].

Based on all the above-mentioned studies, we hypothesize that the distinct sleep disorders during distinct PD neuropathology could be the useful biomarkers of onset and progression follow-up of the neurodegenerative processes in humans, and that the hippocampal GABAergic system plays an important pathophysiological role. Therefore, we aimed this study to further investigate the local sleep disorders from functional (sleep architecture, EEG microstructure of all sleep states, all sleep states episode dynamics and their EEG oscillations, sleep spindles, locomotor activity, and spatial memory abilities) to cellular (local GABAergic interneurons) and molecular (microtubule-associated protein 2 (MAP2) and postsynaptic density protein 95 (PSD-95)) levels in rats following the bilateral PPT lesion (rats with PD cholinopathy), unilateral SNpc lesion (hemiparkinsonian rats), and combined unilateral SNpc/bilateral PPT lesion (hemiparkinsonian rats with PD cholinopathy).

## 2. Results

### 2.1. Alterations of the Hippocampal PV+ Interneurons in the Distinct Rat Models of PD Neuropathology

In this study, in order to investigate the cellular substrate of the local (hippocampal) prodromal sleep disorders in the distinct rat models of PD neuropathology, we quantified the PV+ interneurons of dentate gyrus (DG) by using the hippocampal sections from our previous study, where we have already verified and quantified the lesions: the bilateral PPT lesion (PD cholinopathy), the unilateral SNpc lesion (hemiparkinsonism), and the combined bilateral PPT/unilateral SNpc lesion (hemiparkinsonism with PD cholinopathy) [33].

As previously reported, the PPT cholinergic deficit in the bilaterally PPT lesioned rats (the rat model of PD cholinopathy) was higher than 20% throughout the overall PPT antero-posterior dimension, and the dopaminergic deficit in the SNpc lesioned rats (the rat model of hemiparkinsonism) was higher than 60% throughout the overall SNpc antero-posterior dimension [33]. Furthermore, in the SNpc/PPT lesioned rats (the rat model of hemiparkinsonism with PD cholinopathy), the PPT cholinergic deficit was the same as in the model of PD cholinopathy, except for the small difference in the middle of this structure, and the SNpc dopaminergic deficit was higher than 48% throughout the overall SNpc antero-posterior dimension, with the highest deficit posteriorly (>72%) [33].

In our present study, we quantified the number of PV+ interneurons of the DG throughout the overall hippocampal antero-posterior dimension per each brain side of each rat in each experimental group. We used three defined stereotaxic ranges: −1.50–3.00 mm; −3.10–4.60 mm; and −4.70–6.20 mm posterior from bregma (Figure 1 and Figure 2).

While there was no alteration in the number of PV+ interneurons within DG in the hemiparkinsonian rats (Figure 1, SNpc lesion; *n* = 4; z ≥ −1.01, *p* ≥ 0.34) versus the controls (Control; *n* = 6), there was a significantly reduced number of PV+ interneurons in DG throughout the overall antero-posterior hippocampal dimension in the hemiparkinsonian rats with PD cholinopathy (Figure 1, SNpc/PPT lesion; *n* = 8; z ≥ −2.71, *p* ≤ 0.02). In contrast, we evidenced the increased number of PV+ interneurons in DG from −1.50 to −4.60 mm posterior from bregma in the rats with PD cholinopathy (Figure 1, PPT lesion; *n* = 4; z ≥ −2.05, *p* = 0.04). The typical individual examples of PV immunostaining of each experimental group versus control, and throughout the overall DG antero-posterior dimension, are depicted in Figure 2.

Furthermore, we correlated the number of PV+ interneurons in DG with the SNpc dopaminergic or PPT cholinergic deficits in each rat model of PD (Figure 3). In contrast to the SNpc dopaminergic deficit that showed no correlation with the number of hippocampal PV+ interneurons in the hemiparkinsonian rats (Figure 3B, r = 0.32, *p* = 0.20 for the SNpc lesion (*n* = 4); r = −0.10, *p* = 0.69 for the SNpc/PPT lesion (*n* = 7)), the PPT cholinergic deficit was positively correlated with the number of hippocampal PV+ interneurons in the rats with PD cholinopathy (Figure 3A, PPT lesion (*n* = 4); r = 0.52, *p* = 10^−5^), as well as in the hemiparkinsonian rats with PD cholinopathy (Figure 3A, SNpc/PPT lesion (*n* = 6); r = 0.29, *p* = 0.05).

### 2.2. Impact of the Hippocampal PV+ Interneurons on Local Sleep Architecture and NREM/REM EEG Microstructure in the Distinct Rat Models of PD Neuropathology

In our previous study, we demonstrated the hippocampal sleep fragmentation (increased number of Wake episodes) in the rat models of hemiparkisonism and hemiparkinsonism with PD cholinopathy [33].

In this study, we did not find any functional coupling (correlation) between the number of the PV+ interneurons in DG and the hippocampal Wake/NREM/REM state duration or their episode number/duration in the control rats (*p* ≥ 0.32; data). On the other hand, we demonstrated the functional coupling between the altered number of PV+ interneurons in DG and Wake episodes duration only in the hemiparkinsonian rats with PD cholinopathy (Figure 4, SNpc/PPT lesion (*n* = 7); r = 0.52, *p* = 0.01) in contrast to all other experimental groups (Figure 4, Control (*n* = 3); PPT lesion (*n* = 3); SNpc lesion, (*n* = 3); r ≥ −0.25, *p* ≥ 0.36). Namely, as much as the number of PV+ interneurons was reduced in DG of the hemiparkinsonian rats with PD cholinopathy (SNpc/PPT lesion), the duration of hippocampal Wake episodes was shorter (Figure 4).

Furthermore, we did not evidence any functional coupling between the number of the PV+ interneurons in DG and the hippocampal NREM/REM delta, theta, sigma, beta and gamma relative amplitudes, neither in control rats (*p* ≥ 0.60 for NREM; *p* ≥ 0.33 for REM), nor in any PD model (*p* ≥ 0.37 for NREM; *p* ≥ 0.37 for REM).

### 2.3. Sleep Spindle Dynamics in the Distinct Rat Models of PD Neuropathology

Our present results demonstrate that the sleep spindles (SSs) occurred during NREM and REM sleep in control rats only within the motor cortex (Table 1 and Table 2, Control).

PD cholinopathy induced SSs occurrence during NREM sleep within the hippocampus (Table 1, PPT lesion), as well as HVSs occurrence, both within the motor cortex and hippocampus during NREM and REM sleep (Table 2, PPT lesion). These induced motor cortical and hippocampal HVSs were more dense and longer during REM sleep (Figure 5A, Table 2, PPT lesion; z ≥ −5.99, *p* ≤ 10^−3^ for the motor cortex; z ≥ −3.69, *p* ≤ 10^−3^ for the hippocampus). Conversely, the motor cortical SSs became longer but sparse during REM sleep (Table 1, PPT lesion; z ≥ −2.42, *p* = 0.02).

Hemiparkinsonism induced the SSs occurrence during hippocampal NREM sleep (Table 1, SNpc lesion), but it induced the HVSs occurrence only during REM sleep within both the motor cortex and hippocampus (Table 2, SNpc lesion).

Hemiparkinsonism with PD cholinopathy induced SSs occurrence during hippocampal REM sleep (Table 1, SNpc/PPT lesion), HVSs occurrence during both NREM and REM sleep in the motor cortex, and also HVSs occurrence only during REM sleep in the hippocampus (Table 2, SNpc/PPT lesion).

SSs and HVSs are always slower oscillations within the hippocampus than in the motor cortex (Table 1 and Table 2; z ≥ −3.04, *p* ≤ 0.02), apart from hemiparkinsonism with PD cholinopathy (Table 1 and Table 2; z = −1.69, *p* = 0.09). Particularly, the inter-structure difference of mean HVS frequency during REM sleep was abolished in the hemiparkinsonian rats with PD cholinopathy, which was due to an increased HVS frequency in the hippocampus (Figure 5B, Table 2; z = −3.92, *p* = 10^−4^).

Our results provide evidence for REM sleep as a predisposing state for HVSs induction in all experimental models of PD, particularly in the hemiparkinsonian models (SNpc lesion and SNpc/PPT lesion). Moreover, PD cholinopathy prolongs HVS and SS duration and increases the density of induced HVSs, particularly during REM sleep (Table 2, Figure 5A).

Correlations between the parameters of HVS dynamic (nHVS, HVS Density/h (1/min), HVSdur/h (min), HVSdur (s), HVSf (Hz)), which were dominantly induced during hippocampal REM sleep in all experimental models of PD, with the number of PV+ interneurons in the DG, did not show any functional coupling (*p* ≥ 0.56), apart from a tendency for the hippocampal HVS duration shortening, due to the reduced number of PV+ interneurons in the DG of the hemiparkinsonian rats with PD cholinopathy (r = 0.35, *p* = 0.09).

### 2.4. Impact of the Hippocampal PV+ Interneurons on the Spatial Memory Abilities (Habitual Response) in the Distinct Rat Models of PD

We previously evidenced a lack of a physiological habitual response (the impaired spatial memory abilities) in both groups of the hemiparkinsonian rats, but not in the rats with PD cholinopathy [33]. Here, in this study, we correlated the number of PV+ interneurons in DG with locomotor activity (distance traveled during habitual response) in all experimental groups (Figure 6).

In contrast to the PD cholinopathy and hemiparkinsonism, as well as the control condition, where we did not evidence the correlation of PV+ interneurons and physiological habitual response (Figure 6, Control, (*n* = 6); r = −0.08, *p* = 0.40; PPT lesion, (*n* = 4); r = 0.05, *p* = 0.69) or pathological habitual response (an impaired spatial memory abilities; Figure 6, SNpc lesion, (*n* = 4); r = −0.01, *p* = 0.96), there was the positive correlation between the number of PV+ interneurons and habitual response only in the hemiparkinsonian rats with PD cholinopathy (Figure 6, SNpc/PPT lesion, (*n* = 8); r = 0.36, *p* = 10^−5^). Since these rats have a reduced mean number of PV+ interneurons (see Figure 1), and their spatial memory abilities are impaired (increased locomotor activity over three consecutive days versus the physiological habitual response or decreased locomotor activity over three days; see Petrovic et al. [33]), this positive correlation only indicates a possible important role of the hippocampal PV+ interneurons in the impaired spatial memory abilities in this experimental group of rats.

### 2.5. Hippocampal Synaptic Re-Organization and the PV+ Interneurons Alteration in the Distinct Rat Models of PD Neuropathology

To further explore the synaptic re-organization and the alteration of hippocampal PV+ interneurons, we investigated the hippocampal MAP2 and PSD-95 expression—but only in those experimental groups of rats with the opposite effect on PV+ interneurons expression in DG. Therefore, the hippocampal MAP2 and PSD-95 expression was followed in the control rats, the rats with PD cholinopathy, and the hemiparkinsonian rats with PD cholinopathy.

#### 2.5.1. Hippocampal PV+ Interneurons Alteration and MAP2 Expression in the Distinct Rat Models of PD Neuropathology

Our results show that the PD cholinopathy increased the number of PV+ interneurons within DG and suppressed MAP2 expression, particularly in the DG granular and polymorphic cells layers, and in the pyramidal cell layer and stratum radiatum of the hippocampal CA3 region (Appendix A, Figure 7 and Figure 8, PPT lesion). Conversely, the hemiparkinsonism with PD cholinopathy reduced the number of PV+ interneurons within DG and enhanced MAP2 expression in the DG granular and polymorphic cell layers, and in the pyramidal cell layer and stratum radiatum of the hippocampal CA3 region (Appendix A, Figure 7 and Figure 8, SNpc/PPT lesion).

#### 2.5.2. Hippocampal PV+ Interneurons Alteration and PSD-95 Expression in the Distinct Rat Models of PD Neuropathology

In addition to suppressed MAP2 expression and an increased number of PV+ interneurons within DG, the PD cholinopathy also suppressed the overall hippocampal PSD-95 expression. The suppression of PSD-95 immunoreactivity was particularly evidenced in the granular cell layer and molecular layer of DG, as well as in the pyramidal cell layer and stratum radiatum of the hippocampal CA3 region (Appendix A, Figure 9 and Figure 10, PPT lesion). On the other hand, the hemiparkinsonism with PD cholinopathy reduced the number of PV+ interneurons in DG and enhanced PSD-95 expression—but only within the granular cell layer and molecular layer of DG (Appendix A, Figure 9 and Figure 10, SNpc/PPT lesion).

### 2.6. Alteration of the PV+ Interneurons in the Reticulo-Thalamic Nucleus (RT) in the Distinct Rat Models of PD Neuropathology

In order to investigate the cellular substrate for the induced and distinctly altered sleep spindle dynamics in the distinct models of PD neuropathology, particularly the HVSs dynamic during hippocampal REM sleep, we also followed the alteration of PV+ interneurons of RT, which are important in sleep spindle generation.

We demonstrate the suppression of PV+ interneurons expression in RT of the hemiparkinsonian rats with PD cholinopathy, as well as in the hippocampal DG (Appendix A, SNpc/PPT lesion). Although we did not quantify the number of PV+ interneurons of RT, besides a suppression of the PV+ interneurons expression in almost all the SNpc/PPT lesioned rats (*n* = 7/8), there were also the obvious defects in the dorsal or ventral part of RT (*n* = 3/7), immunostained with PV, on the brain side ipsilateral to the combined SNpc and PPT lesions (always the right brain side) vs. the contralateral brain side, the control rats, the bilateral PPT lesioned rats, and the SNpc lesioned rats (Appendix A). The overall antero-posterior dimensions of these defects in PV immunostainings were from 0.64 to 1.84 mm.

Figure 11 depicts the typical individual examples of RT PV immunostaining per each experimental group of rats (Figure 11A), with the typical example of an overall antero-posterior dimensions of the dorsal and ventral defects within RT (Figure 11Bb, Bd, and Bf) vs. the contralateral RT (Figure 11Ba, Bc, and Be). In this typical example of the suppressed PV+ interneurons expression within RT in the hemiparkinsonian rat with PD cholinopathy, the defect of PV immunostaining was spread from −2.07 to −3.91 mm posterior from bregma (overall the antero-posterior defect of PV immunostaining was 1.84 mm).

### 2.7. Synaptic Re-Organization of RT and the PV+ Interneurons Alteration in the Distinct Rat Models of PD Neuropathology

In parallel to the hippocampal synaptic re-organization, we also investigated MAP2 and PSD-95 expression within RT in the control rats, the rats with PD cholinopathy, and the hemiparkinsonian rats with PD cholinopathy.

#### 2.7.1. PV+ Interneurons Alteration of RT and MAP2 Expression in the Distinct Rat Models of PD Neuropathology

We evidenced that the hemiparkinsonism with PD cholinopathy reduced PV+ interneurons expression within RT and potentiated MAP2 expression, ipsilaterally to the combined SNpc and PPT lesions (the right brain side) likewise in the hippocampus, and versus the RT of contralateral brain side (left side with only the PPT lesion), as well as versus the control rats, and the bilaterally PPT lesioned rats (Appendix A, SNpc/PPT lesion). In addition, MAP2 expression was partially enhanced in the dorsal part of RT in the rats with PD cholinopathy (Appendix A, PPT lesion). The typical examples of MAP2 expression within the RT of the experimental groups of rats with the opposite effect on PV+ interneurons expression in the hippocampal DG and RT (PPT lesion and SNpc/PPT lesion) and versus controls (Control) are depicted in Figure 12A.

#### 2.7.2. PV+ Interneurons Alteration of RT and PSD-95 Expression in the Distinct Rat Models of PD Neuropathology

Although there was no alteration of the number of PV+ interneurons and MAP2 expression was partially increased (see Appendix A), there was the suppression of PSD-95 expression in RT of the rats with PD cholinopathy (Appendix A, PPT lesion). On the other hand, there was the reduced PV+ interneurons expression within RT of the hemiparkinsonian rats with PD cholinopathy along with MAP2 potentiation, and PSD-95 expression was similar to the control level (Appendix A, SNpc/PPT lesion). Typical examples of PSD-95 expression within RT of the experimental groups of rats with the opposite effect on PV+ interneurons expression in the hippocampal DG and RT (PPT lesion and SNpc/PPT lesion) and versus controls (Control) are depicted in Figure 12B.

## 3. Discussion

Our present study demonstrates the underlying alterations of hippocampal and RT GABAergic PV+ interneurons expression and their distinct synaptic re-organizations during the hippocampal (local) prodromal sleep disorders in the distinct rat models of PD neuropathology. Particularly, whereas the PD cholinopathy enhanced, the hemiparkinsonism with PD cholinopathy reduced the number of hippocampal PV+ interneurons in DG (Figure 1 and Figure 2). This opposite alteration of the hippocampal GABAergic PV+ interneurons expression in DG was induced by the bilateral cholinergic deficit higher than 20% throughout the overall PPT antero-posterior dimension (PD cholinopathy) or by the unilateral SNpc dopaminergic deficit higher than 48% along with the bilateral PPT cholinergic deficit higher than 20% throughout the overall PPT/SNpc antero-posterior dimensions (hemiparkinsonism with PD cholinopathy).

Moreover, while there was no correlation of the SNpc dopaminergic deficits with GABAergic PV+ interneurons expression in the hemiparkinsonian rats, the PPT cholinergic deficit was significantly and positively correlated with the number of GABAergic PV+ interneurons in hippocampal DG of the rats with PD cholinopathy, as well as in the hemiparkinsonian rats with PD cholinopathy (Figure 3). Namely, the higher the PPT cholinergic deficit is, the higher the number of GABAergic PV+ interneurons (Figure 3A).

However, we did not find any functional coupling between the number of the hippocampal PV+ interneurons in DG and the hippocampal Wake/NREM/REM states’ duration or their episode number/duration, and the NREM/REM delta, theta, sigma, beta, and gamma EEG relative amplitudes in control rats, and any model of PD neuropathology. We have only evidenced the functional coupling between the altered number of hippocampal PV+ interneurons in DG and the hippocampal Wake episodes’ duration in the hemiparkinsonian rats with PD cholinopathy (Figure 4). Namely, as much as the number of hippocampal PV+ interneurons of DG is reduced, the duration of the hippocampal Wake episodes is shorter in the hemiparkinsonian rats with PD cholinopathy.

In addition, our study demonstrates for the first time that REM sleep is a predisposing state for the HVSs generation in all experimental models of PD neuropathology, particularly during hippocampal REM sleep in the hemiparkinsonian models. Moreover, PD cholinopathy prolongs both the HVSs and SSs duration and increases the density of the induced HVSs, particularly during REM sleep (Table 1 and Table 2, Figure 5).

However, we did not find any functional coupling between the parameters of HVS dynamics, which were dominantly induced during hippocampal REM sleep in all experimental models of PD, with the number of hippocampal PV+ interneurons. However, the inter-structure differences of the mean HVS frequency during REM sleep (both types of the sleep spindles are always slower oscillations within the hippocampus vs. the motor cortex) were abolished in the hemiparkinsonian rats with PD cholinopathy, which was due to an increased HVS frequency in the hippocampus (Table 2, Figure 5B).

Although we previously demonstrated impaired spatial memory abilities in both rat models of the hemiparkinsonian rats versus the controls and rats with PD cholinopathy [33], our present study demonstrates the significant positive correlation between the number of hippocampal PV+ interneurons in the DG (reduced number of PV+ interneurons) and pathological habitual response (an increased locomotor activity over three consecutive days) only in the hemiparkinsonian rats with PD cholinopathy (Figure 6). However, since we used the habitual response as an indirect measure of the memory abilities (not an appropriate test of memory abilities) and we quantified the PV+ interneurons on .tiff images and only within the overall dimension of DG, this positive correlation may only indicate a possible important role of the hippocampal GABAergic PV+ interneurons in the impaired spatial memory abilities in this experimental group of rats (Figure 6), which needs further investigation.

Furthermore, our results demonstrate the opposite alteration of the hippocampal GABAergic PV+ interneurons expression of DG in the PD cholinopathy vs. the hemiparkinsonism with PD cholinopathy along with the distinct local (hippocampal) and remote (RT) MAP2 and PSD-95 expressions. While the PD cholinopathy enhanced hippocampal PV+ interneurons expression in DG and suppressed the hippocampal MAP2 and PSD-95 expression, the hemiparkinsonism with PD cholinopathy reduced hippocampal PV+ interneurons expression of DG and induced an overexpression of the hippocampal MAP2 and PSD-95 (Figure 7, Figure 8, Figure 9 and Figure 10, Appendix A). In addition to the locally enhanced PV+ interneurons expression and MAP2/PSD-95 suppression in the hippocampus, there was no alteration of PV+ interneurons expression within RT, and there was a partial remote enhancement of MAP2 expression only in the dorsal part of RT along with PSD-95 suppression during PD cholinopathy (Figure 12 and Appendix A). Conversely, there was the reduced hippocampal/RT number of PV+ interneurons in the hemiparkinsonian rats with PD cholinopathy along with the enhanced hippocampal/RT MAP2 expression, and there was enhanced PSD-95 expression only in the hippocampus (Figure 11 and Figure 12, and Appendix A).

In our present study, we evidenced the opposite level of the parvalbumin, pre-synaptic protein MAP2, and postsynaptic excitatory protein PSD-95 expressions locally in the hippocampus, which could be the underlying mechanisms of distinct hippocampal prodromal sleep disorders in the PD cholinopathy vs. the hemiparkinsonism with PD cholinopathy. On the other hand, the suppression of excitation (detected by the lack of excitatory synaptic protein PSD-95 expression) in the RT of the rats with PD cholinopathy vs. the hemiparkinsonian rats with PD cholinopathy indicates the important PPT cholinergic afferent system and the parvalbumin GABA neurons regulatory role in RT. The lack of excitation in RT (no PV+ interneurons expression change along with suppressed PSD95 expression in RT) of the rats with PD cholinopathy vs. hemiparkinsonian rats with PD cholinopathy could be a reason for the prolongation of both the HVSs and SSs duration and an increased density of the induced HVSs, particularly during REM sleep (Table 1 and Table 2, Figure 5).

According to the new GABA collapse hypothesis [2], PD is the multisystemic neurodegenerative disease whose clinical symptoms reflect the localization and progression of the most advanced GABA pathology. In addition, the hippocampal GABA PV-expressing interneurons coordinate the hippocampal network dynamics required for memory consolidation [35]. PV+ interneurons are a major type of the GABAergic inhibitory interneurons in the brain, which are characterized by their short action potential duration and their ability to fire at high frequencies [36]. They have multiple dendrites receiving inputs from diverse afferent pathways as well as the numerous perisomatic boutons onto excitatory neurons, together resulting in an integrated feedforward and feedback inhibitory control of both the local circuitry and remote neuronal networks [36,37,38]. They play a crucial role in determining oscillatory network activity and in regulating plasticity following behavioral learning [36], and they are of crucial importance in spatial memory consolidation in the hippocampus [35,36]. Recent study suggests that immediately following learning, the hippocampal PV+ interneurons drive local oscillations and the reactivation of local neuronal populations, which directly promotes network plasticity and long-term memory formation [35]. PV+ interneurons dysfunction has been linked to several brain diseases that involve memory deficits [36].

Our present results are in accordance with evidence that the prodromal non-motor symptoms of PD are related to the GABAergic system, and that distinct sleep disorders, as the prodromal symptoms of PD, may further accelerate the neurodegenerative processes [2]. Although the sleep disorders in PD have multifactorial etiology, the pathological degeneration of neuronal populations, representing the main sleep regulation centers in the brainstem (such as the PPT) and thalamo-cortical pathways (such as the RT), is probably the most relevant factor. It should be noted here that the GABAergic system is involved in every aspect of sleep regulation, and that intraneuronal ion equilibrium, including the optimal calcium level, can be fully recovered during sleep [39].

Furthermore, it is well known that all brain structures, including the structures related to sleep regulation, are anatomically and neurochemically heterogeneous neuronal populations sharing a common mechanism that controls their activity and metabolism through a complex interaction between the GABA and Ca^2+^-dependent neurotransmission and Ca^2+^-dependent neuronal metabolism [2,39,40]. The Ca^2+^/GABA mechanism stabilizes neuronal activity at the cellular and systemic level [40]. GABA is the main inhibitory neurotransmitter within the central nervous system [2,40]. Synaptic transmission, signal transmission, adaptive adjustments, and memory are Ca^2+^/GABA-related mechanisms [2]. The GABA system protects the neurons by the control of calcium influx directly via GABA receptors or indirectly via astrocytes and glial networks [2]. There is evidence that the excessive neuronal activity is firstly tuned by increased GABA inhibition and then further, if necessary, by the reduction of GABA synaptic receptors and calcium channels [2,40]. Therefore, the GABA interneurons are the homeostatic regulators of synaptic inhibition within the cellular networks, and GABA decline etiology appears to apply to all human neurodegenerative processes initiated by abnormal intracellular calcium levels.

The reduction of PV-expressing hippocampal GABA interneurons was reported in several mouse models of autism with an evidenced excitation/inhibition balance shifted toward enhanced inhibition, without GABA neuronal loss, but with PV downregulation [41]. In addition, in the rat model of depression, the PV expression in GABAergic interneurons was reduced in all regions of the hippocampus [42]. Moreover, there is evidence that a reduced PV expression (low levels of the Ca^2+^ binding protein—parvalbumin), via reduction of calcium-buffering capacity, may increase the vulnerability to excitotoxicity [42], as well as an overexpressing of the PV neurons were particularly resistant to excitotoxicity and cell death [42].

Our results show that the PD cholinopathy induced an overexpression of the PV+ interneurons in the hippocampal DG but did not alter the PV+ interneurons expression remotely in RT. Conversely, the hemiparkinsonism with PD cholinopathy reduced the PV+ interneurons expression in hippocampal DG and RT, along with enhanced MAP2 expression in both brain structures, and hippocampal DG enhanced/RT no changed PSD-95 expression, suggesting severe presynaptic and postsynaptic re-organizations at the hippocampal and thalamic level. Therefore, at the level of local field potential, such as sleep spindle, the increased mean intrinsic frequency of hippocampal HVS during REM sleep in the hemiparkinsonian rats with PD cholinopathy (Figure 5B) could be a consequence of PV+ interneurons reduction and/or presynaptic and postsynaptic re-organization in the RT. Moreover, our present study suggests a possible protective role of the hippocampal PV overexpression on the synaptic re-organization in the local hippocampal network, as well as remotely in the RT during the PD cholinopathy, conversely to the reduced hippocampal/RT PV expression during hemiparkinsonism with PD cholinopathy, which induced MAP2/PSD-95 hippocampal overexpression, along with MAP2 overexpression/no changed PSD-95 expression, remotely in the RT. Our results imply an important regulatory role of the PPT cholinergic and the SNpc dopaminergic afferent system on the hippocampal and RT synaptic re-organizations through GABAergic PV+ interneurons.

It should be noted here that RT is a thin sheet of GABAergic neurons chemically and electrically interconnected [43], which surrounds other thalamic nuclei and has a key role in sleep rhythm generation, particularly in sleep spindles generation, but also in delta and slow oscillations [44,45]. In addition to the anatomical, morphological, and neurochemical heterogeneity of RT and its crucial implication in sleep rhythm, the RT was recently implicated in the regulation of local sleep heterogeneity through parallel thalamo-cortical loops [45]. Namely, the RT is strongly innervated by cortical inputs and is a part of reciprocally connected and focalized thalamo-cortical loops. Therefore, the cortical activity could drive RT, which would in return influence the cortex in a heterogeneous local manner [45]. The origin of cortical and subcortical afferents and the thalamic target define the anatomical subregions of RT: while the postero-dorsal part of RT is involved in visual and somatosensory modalities, the anterior part of RT is involved in motor and limbic structures. RT is topographically segregated into different parts with different cellular properties that tune the type of local sleep patterns and local sleep oscillations through distinct thalamo-cortical loops. Therefore, the sleep rhythm or sleep oscillations abnormalities in distinct diseases could be related to altered activity in the local parts of RT. For example, a strong deficit in sleep spindles possibly arises due to impaired RT activity [44,45].

Our present results suggest that distinctly altered GABAergic PV+ interneurons along with a synaptic re-organization in the RT local network could be the underlying mechanisms of HVS generation, particularly during REM sleep, as well as of distinct HVS dynamics in the distinct rat models of PD neuropathology. Our results demonstrate the reduced PV+ interneurons/enhanced MAP2/no change of PSD-95 expression throughout all of the topographically determined RT functional subregions.

In addition, there is evidence that the alteration of an inhibitory transmission in the hippocampus, in particular by the PV+ interneurons, is linked to the spatial memory deficits, and that early treatment of PV interneuron hyperactivity might be clinically relevant in preventing memory decline, local network hyperexcitability, and delaying a progression of neurodegenerative disease such as Alzheimer’s disease [36]. Furthermore, recent study by using in vivo Ca^2+^ imaging and optogenetic evidenced that the activity of DG adult-born neurons during REM sleep is necessary for memory consolidation [46].

Our present results related to the reduced PV+ interneurons expression alongside MAP2 overexpression and PSD-95 overexpression/no change in the hippocampus and RT in hemiparkinsonian rats with PD cholinopathy are in accordance with recent evidence that the dysfunction of GABAergic inhibition and a consequent imbalance between excitation and inhibition result in hyperexcitability and the desynchronization of neuronal networks, leading to impairment of information processing, learning, and memory formation [47].

It should be noted here that a lot of evidence has implicated the calcium-related homeostatic mechanisms, giving rise to the Ca^2+^ hypothesis of brain aging and cell death [2]. Although the oxidative stress and calcium-induced excitotoxicity were considered as important pathophysiological mechanisms leading to neural cell death in PD, still, the factors that make the certain neurons vulnerable to neurodegeneration are unknown [2].

Our study demonstrates for the first time an important regulatory role of the hippocampal and RT GABAergic PV+ interneurons and the synaptic protein dynamic alterations in distinct rat models of PD neuropathology, which are reflected prodromally, distinctly, and long-lasting at the functional level: from distinct local sleep disorders through to the distinct alteration of sleep-related EEG oscillations to distinct alteration of the sleep spindles dynamics. Our results in the rat models of PD neuropathology indicate that augmenting the GABAergic signaling via PV+ interneuron modulation can be effective in improving or ameliorating prodromal sleep disorders and memory deficits in PD.

## 4. Materials and Methods

### 4.1. Experimental Design

We used 31 adult male Wistar rats (each two and a half months old, weighing between 250 and 290 g), which were chronically implanted for sleep recording. The rats were randomly divided into four experimental groups: control rats (implanted controls, *n* = 8), rats with PD cholinopathy (a bilateral PPT lesion group, *n* = 8), hemiparkinsonian rats (a unilateral SNpc lesion group, *n* = 7), and hemiparkinsonian rats with PD cholinopathy (a unilateral SNpc/bilateral PPT lesion group, *n* = 8).

After the surgery and throughout the experimental protocol, the animals were individually housed in custom-made clear plexiglass cages (30 × 30 × 30 cm) on a 12 h light–dark cycle (7 a.m. lights on, 7 p.m. lights off) at 25 °C with food and water ad libitum.

All the procedures were performed in accordance with EEC Directive (2010/63/EU) on the Protection of Animals Used for Experimental and other Scientific Purposes, and the protocol was approved by the Ethics Committee for the Protection of Welfare of Experimental Animals of the Institute for Biological Research “Siniša Stanković”—National Institute of Republic of Serbia, University of Belgrade (Approval No. 01–1490; 28/09/2020), and by the Veterinary Directorate, Department of Animal Welfare, Ministry of Agriculture, Forestry and Water Management of Republic of Serbia (Approval No. 323-07-10509/2020-05/1; 13/10/2020).

### 4.2. Surgical Procedure

The surgical procedures for the chronic electrode implantation for sleep recording have been conducted as previously described [25,30,32,33,34].

In brief, the rats were anesthetized with ketamine/diazepam anesthesia (50 mg/kg, Zoletil^®^ 50, VIRBAC, Carros, France; intraperitoneal injection) and positioned in a stereotaxic frame (Stoelting Co., Dublin, Ireland). We implanted two epidural stainless steel screw electrodes in the motor cortex (MCx; A/P: + 1.0 mm from bregma; R/L: 2.0 mm from the sagittal suture; D/V: 1.0 mm from the skull, in accordance with Paxinos and Watson [48]) and two wire electrodes (stainless-steel teflon-coated wire, Medwire, Mount Vernon, NY, USA) into the CA1 hippocampal regions (Hipp; A/P: −3.6 mm from the bregma; R/L: 2.5 mm from the sagittal suture; D/V: 2.5 mm from the brain surface, in accordance with Paxinos and Watson [48]). To assess skeletal muscle activity (EMG), the bilateral wire electrodes were implanted into the dorsal nuchal musculature, and a stainless-steel screw electrode was implanted in the nasal bone as a ground. All the electrode leads were soldered to a miniature connector plug (39F1401, Newark Electronics, Schaumburg, IL, USA), and the assembly was fixed to the screw electrodes and skull using acrylic dental cement (Biocryl-RN, Galenika a.d. Beograd, Serbia).

All the lesions were performed by stereotaxically guided microinfusions during the same surgical procedure for the implantation of the EEG and EMG electrodes, using a Digital Lab Standard Stereotaxic Instrument (Stoelting Co., Dublin, Ireland) with a Quintessential Stereotaxic Injector (Stoelting Co., Wood Dale, IL, USA) and a Hamilton syringe (10 µL or 1 µL).

PD cholinopathy in Wistar rats was induced by the bilateral PPT lesion, using ibotenic acid (IBO, Sigma-Aldrich, St. Louis, MO, USA). We infused 100 nL of 0.1 M IBO/0.1 M PBS bilaterally into the PPT (A/P: −7.8 mm from the bregma; R/L: 1.9 mm from the sagittal suture; D/V: 7.0 mm from the brain surface, following Paxinos and Watson [48]), as a continuous infusion over 60 s [25,33].

The hemiparkinsonism was induced by the unilateral SNpc lesion, using the 6-hydroxy dopamine hydrobromide salt (6-OHDA, Sigma-Aldrich, St. Louis, MO, USA). We infused 1 µL of 6 µg/µL 6-OHDA, dissolved in ice cold sterile saline (0.9% NaCl), and supplemented with 0.2% ascorbic acid, which served as an anti-oxidant, into the right SNpc (A/P: −5.3 mm from the bregma; R: 2.4 mm from the sagittal suture; D/V: 7.4 mm from the brain surface, following Paxinos and Watson [48]). The 6-OHDA microinfusions were performed as a continuous infusion of 200 nL/min, at a constant flow rate, over 5 min [32,33]. In order to minimize the uptake of 6-OHDA by the noradrenergic neurons, 30 min prior to the microinfusion, each rat received a bolus of desipramine hydrochloride (28.42 mg/kg, i.p., Sigma-Aldrich, Taufkirchen, Germany; pH = 7.4).

To induce hemiparkinsonism with PD cholinopathy, we performed double lesioning, in this case both a unilateral SNpc lesion and a bilateral PPT lesion [33].

After each microinfusion, the needle remained within the local brain tissue for 5 min, allowing the solution to diffuse within the PPT or SNpc. For the bilateral PPT lesions, the Hamilton syringe needle was always washed out following the first IBO microinfusion, before the microinfusion into the contralateral PPT. 

At the end of the surgical procedure, the scalp wounds were sutured, and the rats were allowed to recover for two weeks.

### 4.3. Recording Procedure

All the sleep recording sessions were performed 14 days after the surgical procedure. Sleep was recorded for 6 h during the light phase, starting at 9 a.m. The EEG and EMG activities were differentially recorded. After conventional amplification and filtering (0.3–100 Hz band pass; A-M System Inc., Model 3600, Carlborg, WA, USA), the analogue data were digitized (at a sampling frequency of 256/s) using DataWave SciWorks Experimenter Version 8.0 (DataWave Technologies, Longmont, CO, USA), and the EEG and EMG activities were displayed on a computer monitor and stored on a disk for further off-line analysis [25,30,32,33,34,49].

### 4.4. Behavioral Assessments

Behavioral assessments were done a week following sleep recordings, during the light phase (starting at 9 a.m.), as previously described [32,33]. Before each test, the animals were allowed to habituate to the experimental room for 30 min. The basal locomotor activity was monitored for 30 min using an Opto-Varimex Auto-Track System (Columbus Instruments, Columbus, OH, USA) and expressed as distance (centimeters) traveled in the open field arena. The spatial habituation test was performed over three consecutive sessions (locomotor activity during 30 min in the open arena) separated by 24 h intervals and served as an indirect measure of the spatial memory abilities.

### 4.5. Tissue Processing for Histology

At the end of all the recordings and behavioral assessments, the rats were sacrificed for histology. All animals were deeply anesthetized with ketamine/diazepam and perfused transcardially, with 0.9% saline, followed by a 4% paraformaldehyde (PFA, Sigma Aldrich, Taufkirchen, Germany) in 0.1 M phosphate-buffered saline (PBS, pH = 7.4) and finally with a 10% sucrose solution in 0.1 M PBS. The brains were removed and immersed in 4% PFA overnight, and then in a 30% sucrose solution for several days. The brains were serially sectioned on the cryostat (Leica, Wetzlar, Germany) into coronal 40 μm-thick sections, and the free-floating sections were stored in a cryoprotective buffer for further use [32,33,50].

#### 4.5.1. Lesion Identification and Quantification

The PPT lesion was identified using NADPH–diaphorase histochemistry and quantified based on the number of NADPH–diaphorase positively stained cells within the PPT [51]. As previously described [25,30,33], the free-floating sections were rinsed in 0.1 M PBS (pH = 7.4) and incubated for 1 h at 37 °C in the staining solution containing β-NADPH reduced tetrasodium salt (Serva, Heidelberg, Germany) and dimethyl sulfoxide (DMSO, Sigma-Aldrich, Taufkirchen, Germany) dissolved in substrate solution. The substrate solution contained nitro blue tetrazolium chloride (NBT, Serva, Heidelberg, Germany) and 5-bromo-4-chloro-3-indolyl phosphate (BCIP, Serva, Heidelberg, Germany) dissolved in the substrate buffer at pH = 9.5 (0.1 M Tris, 100 mM NaCl, 5 mM MgCl2). The background staining induced by the endogenous alkaline phosphate was reduced by 2 mM levamisole (Sigma-Aldrich, Taufkirchen, Germany). Finally, all the sections were mounted on slides, placed in a clearing agent (Xylene, Zorka Pharma, RS), coverslipped using DPX (Sigma-Aldrich, Burlington, MA, USA), and examined under a Zeiss Axiovert microscope with a camera (Zeiss, Jena, Germany).

The SNpc lesion was identified by tyrosine hydroxylase (TH) immunohistochemistry and quantified based on the number of TH immunostained cells within the SNpc. The brain sections were initially thoroughly rinsed with 0.1 M PBS. The endogenous peroxidase activity was neutralized using 3% hydrogen peroxide/10% methanol/0.1 M PBS for 15 min, and non-specific binding was prevented by 60 min of incubation in 5% normal donkey serum (D9663, Sigma-Aldrich, Burlington, MA, USA)/0.1 M PBS at room temperature [32,33]. The sections were further incubated for 48 h at +4 °C with a primary mouse monoclonal anti-TH antibody (dil. 1:16,000, T2928, Sigma-Aldrich, Burlington, MA, USA) in a blocking solution with 0.5% Triton X-100 (Sigma-Aldrich, Burlington, MA, USA), and subsequently for 90 min in polyclonal rabbit anti-mouse immunoglobulin (dil. 1:100, P0260, Agilent Dako, Glostrup, Denmark). Between each immunolabeling step, the sections were washed in fresh 0.1 M PBS (3 × 5 min). The immunoreactive signals were visualized using a diaminobenzidine solution (1% 3,3′–diaminobenzidine (11208, Acros organics, Geel, Belgium)/0.3% hydrogen peroxide/0.1 M PBS). All the sections were finally mounted on slides, dehydrated in a series of increasing ethanol solutions (ethanol 70%, 96%, 100%, Zorka Pharma, Sabac, RS), placed in a clearing agent (Xylene, Zorka Pharma, Sabac, RS), coverslipped with DPX (Sigma-Aldrich, Burlington, USA), and examined under a Leica light microscope with a camera(Leica, Wetzlar Germany). To test the specificity of the immunolabeling, the primary antibodies were omitted in the control experiments.

The quantification of cholinergic and/or dopaminergic neuronal loss was done by counting the NADPH–diaphorase or TH positively stained cells using ImageJ 1.46 software [25,30,32,33]. For this purpose, all the tissue samples of the corresponding experimental group and brain structure (three sections per each rat and each brain structure) were grouped into three stereotaxic ranges defined according to the overall PPT or SNpc antero-posterior dimension (for the SNpc lesion: −4.60–5.10, −5.20–5.70, and −5.80–6.30 mm posterior from the bregma; for the PPT lesion: −6.90–7.40, −7.50–8.00, and −8.10–8.60 mm posterior from the bregma). The neuronal losses were expressed with respect to the mean control absolute number for each sterotaxic range of the PPT/SNpc, which was taken as 100%. The unilateral SNpc lesions were quantified with respect to its corresponding contralateral SNpc, whereas the bilateral PPT lesions were quantified with respect to controls [25,30,32,33].

#### 4.5.2. Immunohistochemistry for PV, MAP2 and PSD-95

The free-floating brain sections were initially thoroughly rinsed with 0.1 M PBS (pH = 7.4). The non-specific binding was prevented by incubation in 3% hydrogen peroxide/10% methanol/0.1 M PBS for 15 min and 5% normal donkey serum/0.1 M PBS (D9663, Sigma-Aldrich, Burlington, MA, USA) for 60 min at room temperature. Then, the sections were incubated overnight at +4 °C with the following primary antibodies: mouse monoclonal anti-PV antibody (dil. 1:2000, P3088, Sigma-Aldrich, Burlington, MA, USA) [50], mouse monoclonal anti-MAP2 antibody (dil. 1:6000, MAB378, Merck Millipore, Burlington, MA, USA), and mouse monoclonal anti-PSD-95 antibody (dil. 1:200, MAB1598, Merck Millipore, Burlington, MA, USA). The primary antibodies were diluted in PBS containing 0.5% Triton X-100 (for anti-PV) or 0.1% Triton X-100 (for anti-MAP2 and anti-PSD-95). After three 5-min washes in 0.1 M PBS, the sections were incubated for 90 min with polyclonal rabbit anti-mouse immunoglobulin (dil. 1:100, P0260, Agilent Dako, Glostrup, Denmark). The immunoreactive signals were visualized using a diaminobenzidine solution (1% 3,3–diaminobenzidine [11208, Acros organics, Geel, Belgium]/0.3% hydrogen peroxide/0.1 M PBS). All the sections were finally mounted on slides, dehydrated through increasing alcohol concentrations (70%, 96%, and 100% ethanol, Zorka Pharma, Sabac, RS), placed in a clearing agent (Xylene, Zorka Pharma, Sabac, RS), coverslipped with DPX (Sigma-Aldrich, Burlington, MA, USA), and examined under a Leica light microscope with a camera (Leica, Wetzlar Germany). To test the specificity of immunostaining, the primary antibodies were omitted in the control experiments.

#### 4.5.3. Quantification of PV Immunostaining

The quantification of PV immunoreactivity within the dentate gyrus (DG) was done by using the ImageJ 1.46 software (NIH, Bethesda, MD, USA,) and counting the number of PV immunoreactive (PV+) interneurons. For this purpose, all the tissue samples of the corresponding experimental groups were grouped into three stereotaxic ranges covering the overall hippocampal antero-posterior dimension. The defined stereotaxic ranges were −1.50–3.00 mm, −3.10–4.60 mm, and −4.70–6.20 mm posterior from bregma. For all experimental groups, the number of PV+ interneurons was counted per each brain side, at each stereotaxic range, pulled for each experimental group, and expressed as the mean number + SE.

### 4.6. Sleep Analysis

The sleep analysis was done in MATLAB R2011a (MathWorks Inc., Natick, MA, USA) using software originally developed in MATLAB 6.5 [25,30,32,33,34,49].

We applied the FFT algorithm to the signals acquired throughout each 6-hour recording (2160 10 s Fourier epochs in total) and automatically differentiated each 10 s epoch as Wake, NREM, or REM state [25,30,32,33,34,49]. To assess the local sleep, we particularly extracted the simultaneous and non-simultaneous Wake/NREM/REM 10-s epochs of the motor cortex and the hippocampus [30,32,33,34] for further analysis of the local sleep architecture (Wake/NREM/REM state duration), local state-related episode dynamics (Wake/NREM/REM episode number and episode duration), and local state-related EEG microstructure (Wake/NREM/REM relative amplitudes of all the conventional EEG frequency bands) [25,30,32,33,34,49]. 

In addition, we have also analyzed the sleep spindle (SS) and high-voltage sleep spindle (HVS) dynamics during 1 h of NREM and REM sleep (extracted always between the 3rd and 4th hour of sleep recording) simultaneously recorded in the motor cortex and hippocampus. The automatic detection of SSs and HVSs was followed by visual validation of all the detected SSs and HVSs before the final extraction and analysis [29,30,32,52]. Namely, after the EEG signals were band-pass filtered (11–17 Hz for SS and 4.1–10 Hz for HVS), we applied the continuous wavelet transform with the mother wavelet “cmorl-2” function, providing a complex Morlet wavelet with a determined central frequency *f*_0_ = 2 [29,30,32,52,53]. Additional detection criteria included a minimum duration set to 0.5 s for SS and 1 s for HVS. However, automatic detection had to be visually corrected, since some detections were false positive, false negative, or inaccurate (oscillation was not detected in the overall duration). For final analysis of spindle dynamics (mean density, mean intrinsic frequency, mean duration per 1 h of NREM and REM sleep), all the visually detected SSs or HVSs were extracted and concatenated for each structure (motor cortex or hippocampus), each state (NREM or REM sleep), and each experimental group (Control, PPT lesion, SNpc lesion, and SNpc/PPT lesion).

### 4.7. Statistical Analysis

All statistical analyses were performed using a Kruskal–Wallis ANOVA (χ^2^ values) with the Mann–Whitney U (z-values) two-tailed post hoc test. The accepted level of significance in all cases was *p* ≤ 0.05. For the correlation analysis, we employed Pearson’s correlation coefficient with the accepted level of significance of *p* ≤ 0.05.

## Figures and Tables

**Figure 1 ijms-22-08922-f001:**
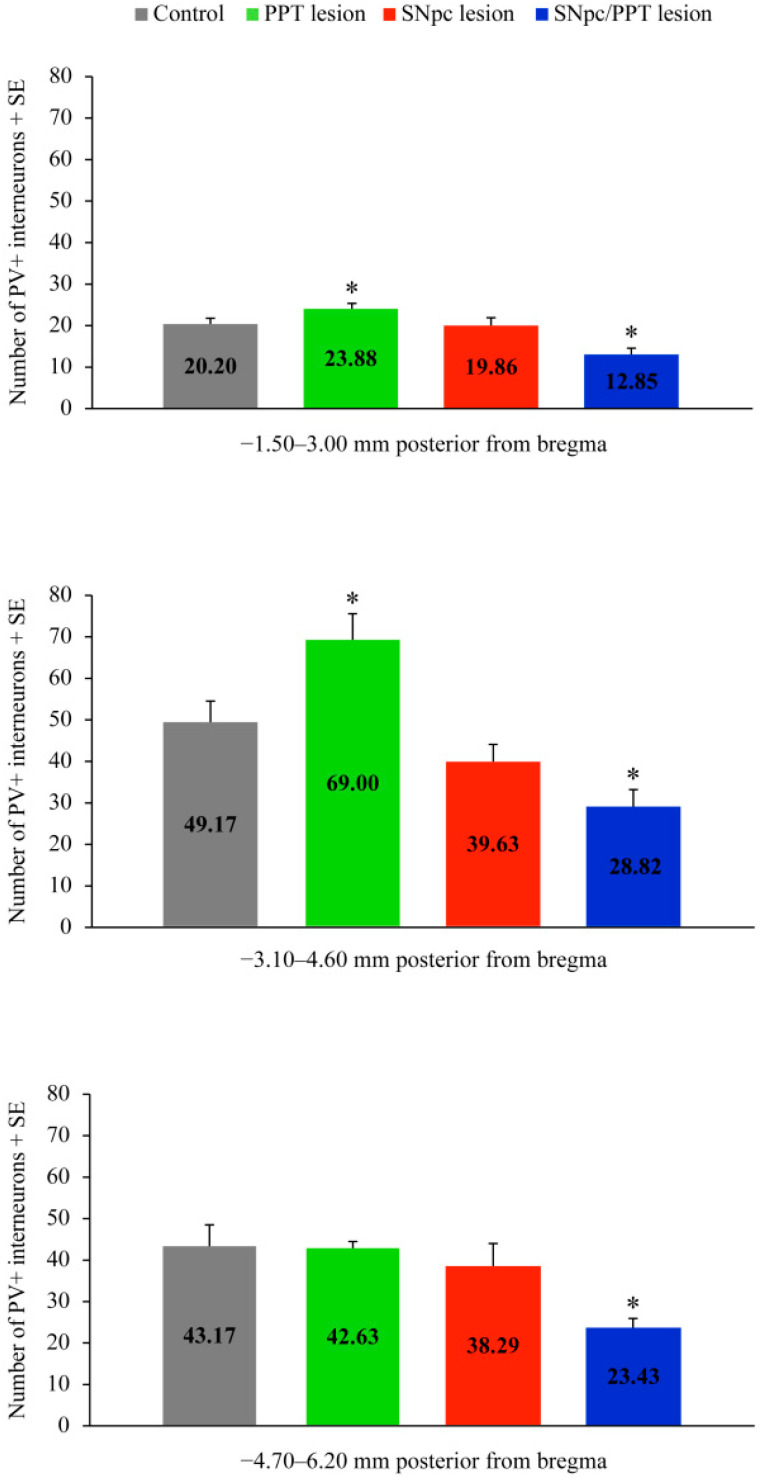
Alterations of the PV+ interneurons within DG in the distinct rat models of PD neuropathology. Group mean numbers of PV+ interneurons in control rats (Control; *n* = 6), rats with PD cholinopaty (PPT lesion; *n* = 4), hemiparkinsonian rats (SNpc lesion; *n* = 4), and hemiparkinsonian rats with PD cholinopathy (SNpc/PPT lesion; *n* = 8) counted in three defined stereotaxic ranges of the overall hippocampal antero-posterior dimension of DG. * indicates the statistically significant difference in mean values at *p* ≤ 0.05 following Mann–Whitney U-test. We evidenced the opposite alterations in the number of PV+ interneurons within DG: the significantly reduced number of PV+ interneurons throughout the overall antero-posterior dimension of DG in the hemiparkinsonian rats with PD cholinopathy (SNpc/PPT lesion) in contrast to the significantly increased number of PV+ interneurons in the rats with PD cholinopathy (PPT lesion).

**Figure 2 ijms-22-08922-f002:**
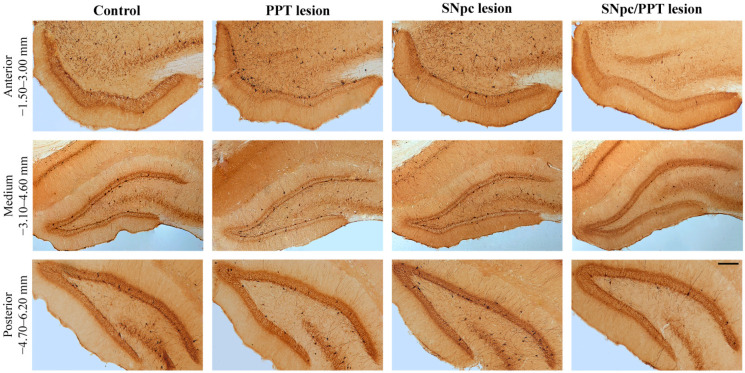
Individual examples of the PV+ interneurons alteration within DG in the distinct rat models of PD neuropathology. The typical examples of PV+ interneurons at three defined stereotaxic ranges throughout the overall antero-posterior dimension of DG (Anterior: −1.50–3.00 mm posterior from bregma; Medium: −3.10–4.60 mm posterior from bregma; Posterior: −4.70–6.20 mm posterior from bregma) in the control rat (Control), the rat with PD cholinopathy (PPT lesion), the hemiparkinsonian rat (SNpc lesion), and the hemiparkinsonian rat with PD cholinopathy (SNpc/PPT lesion). Scale bar is 200 μm.

**Figure 3 ijms-22-08922-f003:**
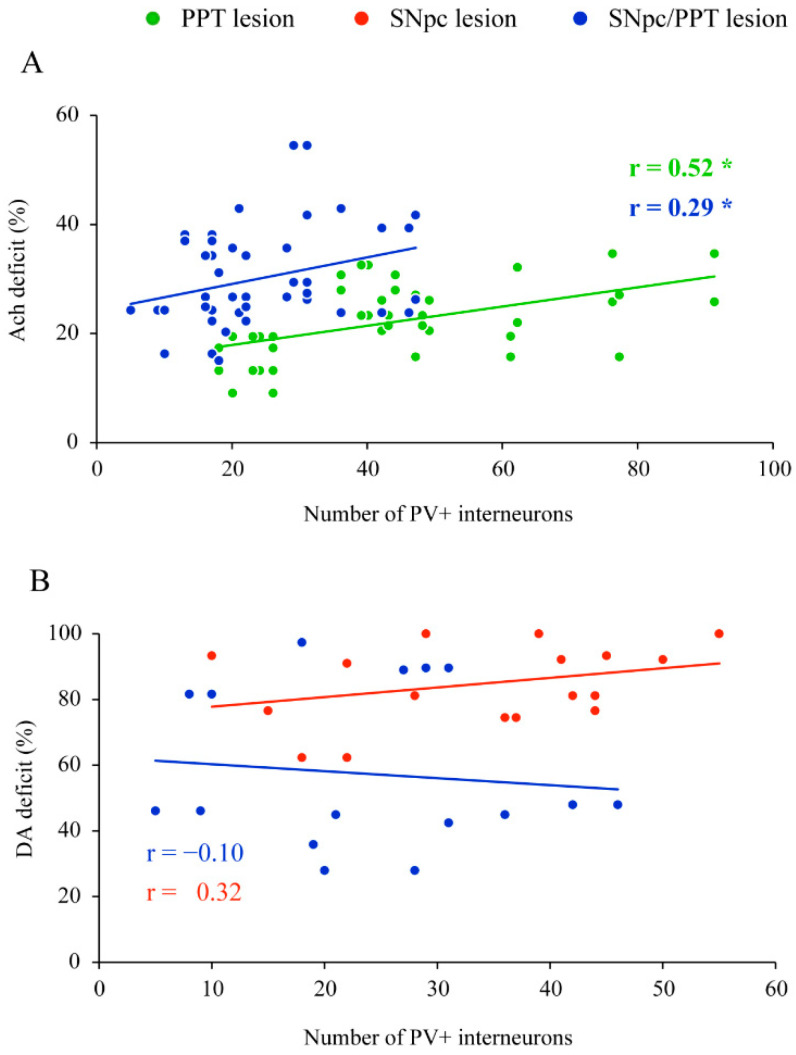
Functional coupling between the altered number of PV+ interneurons within DG and the PPT cholinergic or SNpc dopaminergic deficits in the distinct rat models of PD neuropathology. (**A**) The significant positive correlation between the number of PV+ interneurons in DG and the PPT cholinergic (Ach) deficit in the rats with PD cholinopathy (PPT lesion; *n* = 4) and hemiparkinsonian rats with PD cholinopathy (SNpc/PPT lesion; *n* = 6); (**B**) No significant correlation between the number of PV+ interneurons in DG and SNpc dopaminergic (DA) deficit in the hemiparkinsonian rats (SNpc lesion; *n* = 4) and hemiparkinsonian rats with PD cholinopathy (SNpc/PPT lesion; *n* = 7). Bold numbers with * indicate the statistically significant correlations at *p* ≤ 0.05.

**Figure 4 ijms-22-08922-f004:**
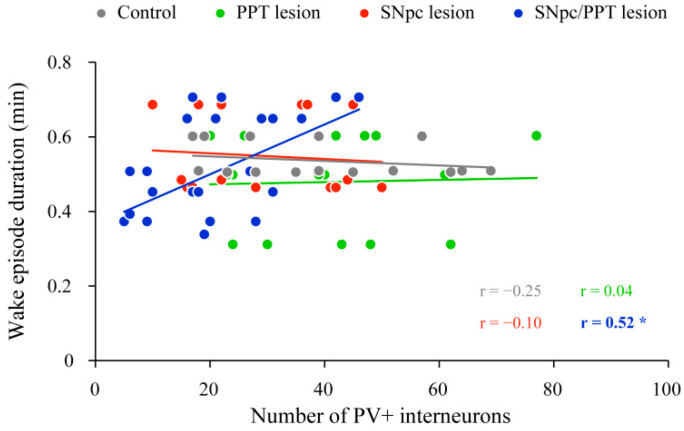
Functional coupling between the altered number of PV+ interneurons within DG and Wake episode duration in the distinct rat models of PD neuropathology. The significant positive correlation between the number of PV+ interneurons in DG and Wake episode duration in the hemiparkinsonian rats PD cholinopathy (SNpc/PPT lesion; *n* = 7) in contrast to no significant correlations in the control rats (Control; *n* = 3), the rats with PD cholinopathy (PPT lesion; *n* = 3) and the hemiparkinsonian rats (SNpc lesion; *n* = 3). Bold numbers with * indicate the statistically significant correlations at *p* ≤ 0.05.

**Figure 5 ijms-22-08922-f005:**
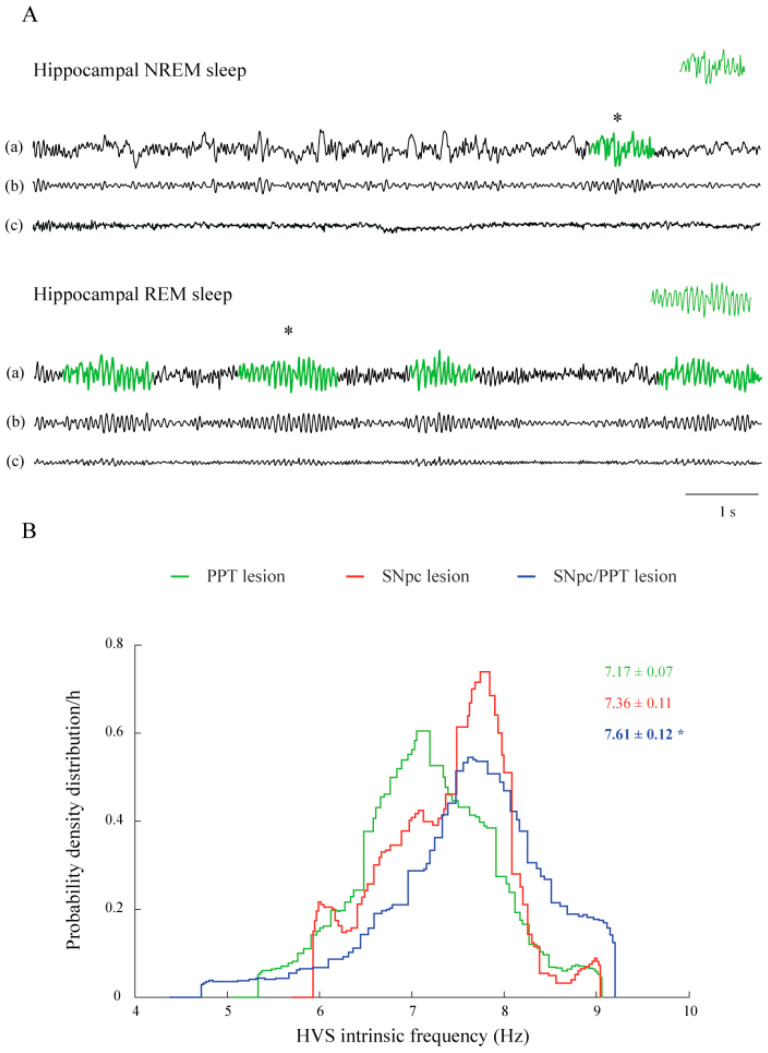
Hippocampal NREM/REM related HVS dynamic in the distinct rat models of PD neuropathology. (**A**) Individual examples of HVSs during hippocampal NREM and REM sleep in the rat with PD cholinopathy (PPT lesion): analog 20 s EEG signals of the hippocampal NREM and REM sleep with detected HVSs marked in green (**a**), corresponding band pass filter (4.1–10 Hz), used for the detection and extraction of the HVSs from the EEG signal (**b**), and corresponding EMG (**c**). The PD cholinopathy induced the prolongation and higher density of hippocampal HVSs during REM sleep. * marks the extracted examples of HVSs during hippocampal NREM and REM sleep (upper right inserts) in the rat model of PD cholinopathy. (**B**) The group probability density distributions of the hippocampal HVS intrinsic frequency during REM sleep in the distinct rat models of PD. The group distributions included an assembly of 104 HVSs from eight rats with PD cholinopathy, 35 HVSs from five hemiparkinsonian rats, and 56 HVSs from eight hemiparkinsonian rats with PD cholinopathy. Bold numbers with * indicate the statistically significant difference of the hippocampal mean HVS intrinsic frequency values during REM sleep at *p* ≤ 0.05 following Mann–Whitney U-test. The hemiparkinsonism with PD cholinopathy increased the hippocampal HVS intrinsic frequency during REM sleep (SNpc/PPT lesion, the right shifted distribution, blue line).

**Figure 6 ijms-22-08922-f006:**
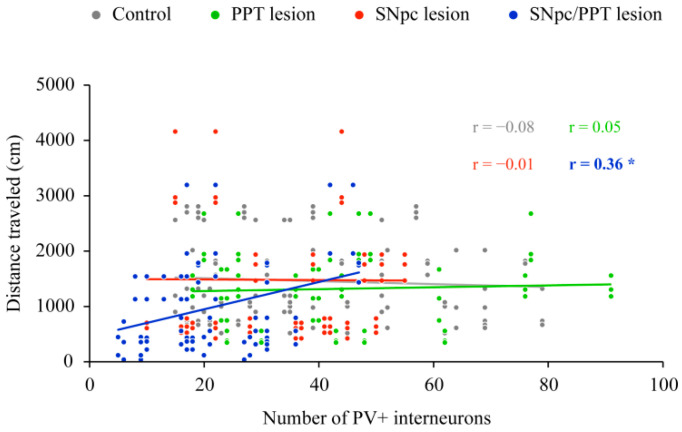
Functional coupling between the altered number of PV+ interneurons within DG and the spatial memory abilities in the distinct rat models of PD neuropathology. The significant positive correlation between the number of PV+ interneurons in DG and distance traveled during habitual response in the hemiparkinsonian rats with PD cholinopathy (SNpc/PPT lesion; *n* = 8) in contrast to no significant correlations in the control rats (Control; *n* = 6), the rats with PD cholinopaty (PPT lesion; *n* = 4) and hemiparkinsonian rats (SNpc lesion; *n* = 4). Bold numbers with * indicate the statistically significant correlations at *p* ≤ 0.05.

**Figure 7 ijms-22-08922-f007:**
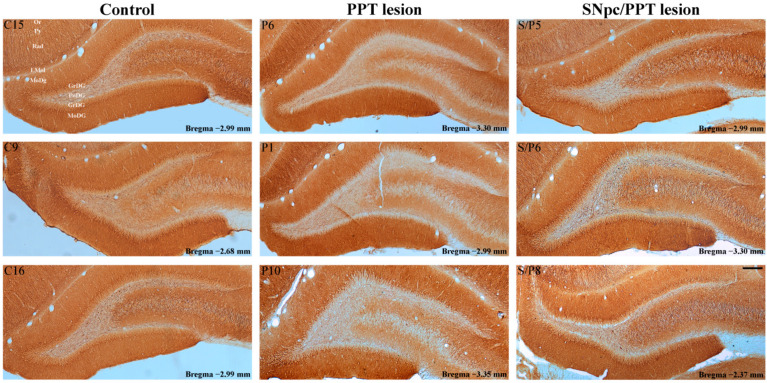
Alteration of MAP2 immunostaining within DG in the distinct rat models of PD neuropathology. MAP2 immunoreactivity within DG of the control rats (Control, C15, C9, C16), the rats with PD cholinopathy (PPT lesion, P6, P1, P10) and the hemiparkinsonian rats with PD cholinopathy (SNpc/PPT lesion, S/P5, S/P6, S/P8). Or—oriens layer of the hippocampus; Py—pyramidal cell layer of the hippocampus; Rad—radiatum layer of the hippocampus; LMol—lacunosum molecular layer of the hippocampus; MoDG—molecular layer of the dentate gyrus; GrDG—granule cell layer of the dentate gyrus; PoDG—polymorph cell layer of the dentate gyrus. Scale bar is 200 μm. MAP2 immunoreactivity was suppressed in the rats with PD cholinopathy, but it was enhanced in the hemiparkinsonian rats with PD cholinopathy within DG granular and polymorphic cell layers vs. the control rats.

**Figure 8 ijms-22-08922-f008:**
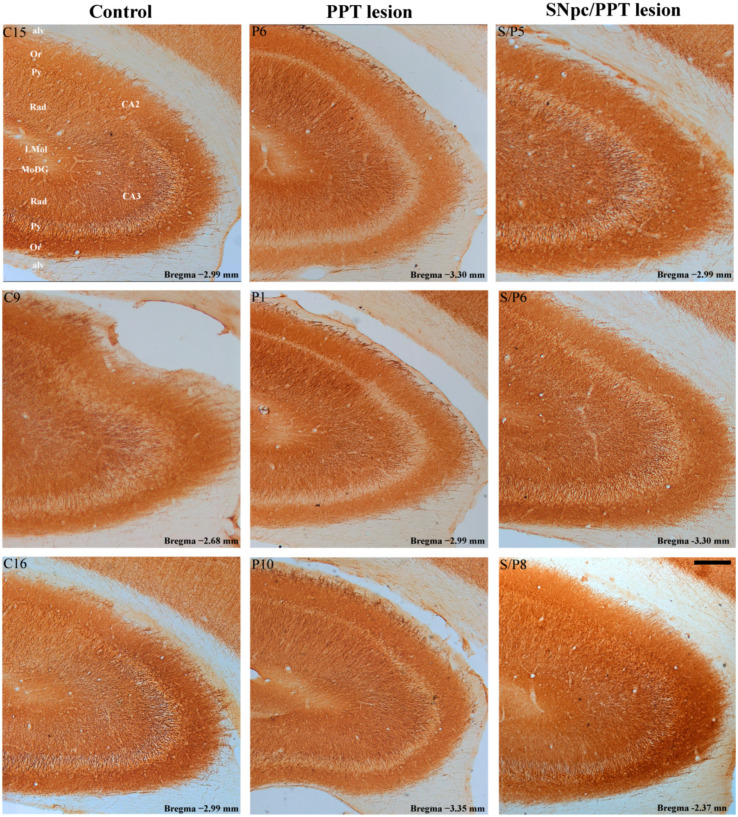
Alteration of MAP2 immunostaining within CA3 in the distinct rat models of PD neuropathology. MAP2 immunoreactivity within CA3 of the control rats (Control, C15, C9, C16), the rats with PD cholinopathy (PPT lesion, P6, P1, P10) and the hemiparkinsonian rats with PD cholinopathy (SNpc/PPT lesion, S/P5, S/P6, S/P8). alv—alveus of the hippocampus; Or—oriens layer of the hippocampus; Py—pyramidal cell layer of the hippocampus; Rad—radiatum layer of the hippocampus; LMol—lacunosum moleculare layer of the hippocampus; MoDG—molecular layer of the dentate gyrus; CA2—CA2 field of the hippocampus; CA3—CA3 field of the hippocampus. Scale bar is 200 μm. MAP2 immunoreactivity was suppressed in the rats with PD cholinopathy, but it was enhanced in the hemiparkinsonian rats with PD cholinopathy within the pyramidal cell layer and stratum radiatum of hippocampal CA3 region vs. the control rats.

**Figure 9 ijms-22-08922-f009:**
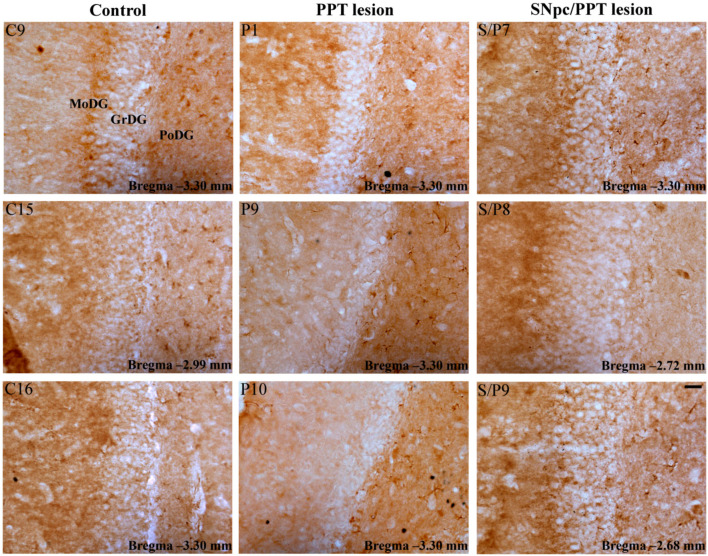
Alteration of PSD-95 immunostaining within the hippocampal DG in the distinct rat models of PD neuropathology. PSD-95 immunoreactivity within DG of the control rats (Control, C9, C15, C16), the rats with PD cholinopathy (PPT lesion, P1, P9, P10), and the hemiparkinsonian rats with PD cholinopathy (SNpc/PPT lesion, S/P7, S/P8, S/P9). MoDG—molecular layer of the dentate gyrus; GrDG—granule cell layer of the dentate gyrus; PoDG—polymorph cell layer of the dentate gyrus. Scale bar is 25 μm. Whereas PSD-95 immunoreactivity was suppressed in the rats with PD cholinopathy, it was enhanced in the hemiparkinsonian rats with PD cholinopathy within the granular cell and molecular layers of DG vs. the control rats.

**Figure 10 ijms-22-08922-f010:**
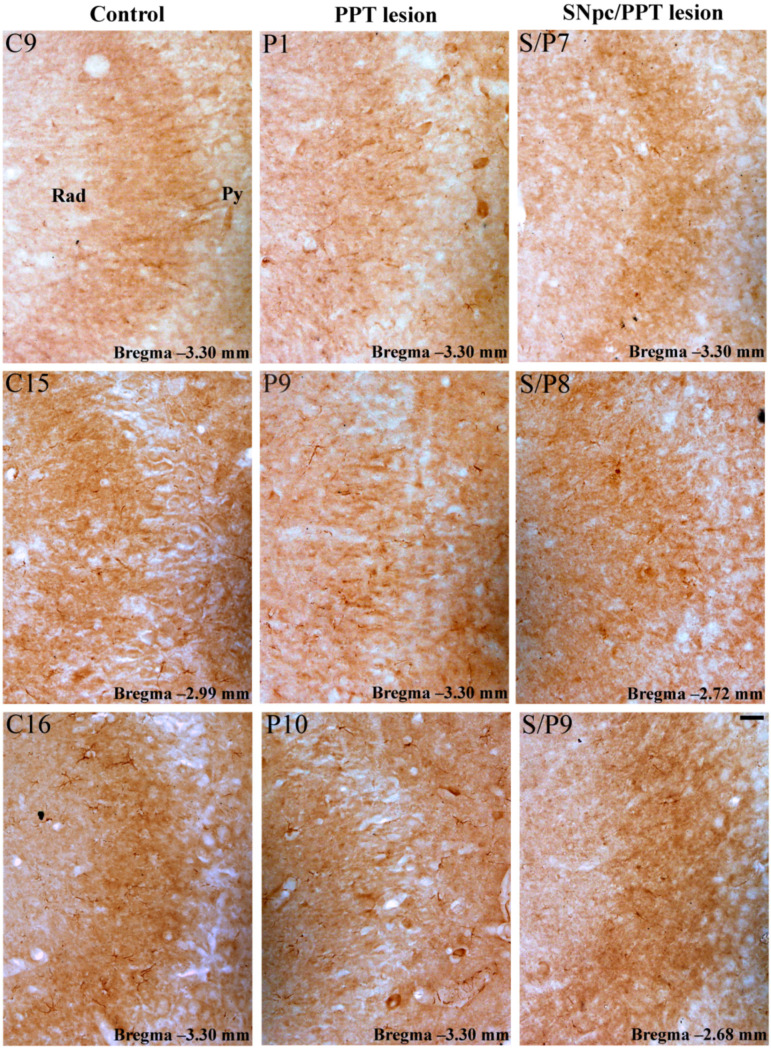
Alteration of PSD-95 immunostaining within the hippocampal CA3 in the distinct rat models of PD neuropathology. PSD-95 immunoreactivity within CA3 of the control rats (Control, C9, C15, C16), the rats with PD cholinopathy (PPT lesion, P1, P9, P10) and the hemiparkinsonian rats with PD cholinopathy (SNpc/PPT lesion, S/P7, S/P8, S/P9). Py—pyramidal cell layer of the hippocampus; Rad—radiatum layer of the hippocampus. Scale bar is 25 μm. PSD-95 immunoreactivity was suppressed in the rats with PD cholinopathy within the pyramidal cell layer and stratum radiatum of the hippocampal CA3 region vs. the control rats and the hemiparkinsonian rats with PD cholinopathy.

**Figure 11 ijms-22-08922-f011:**
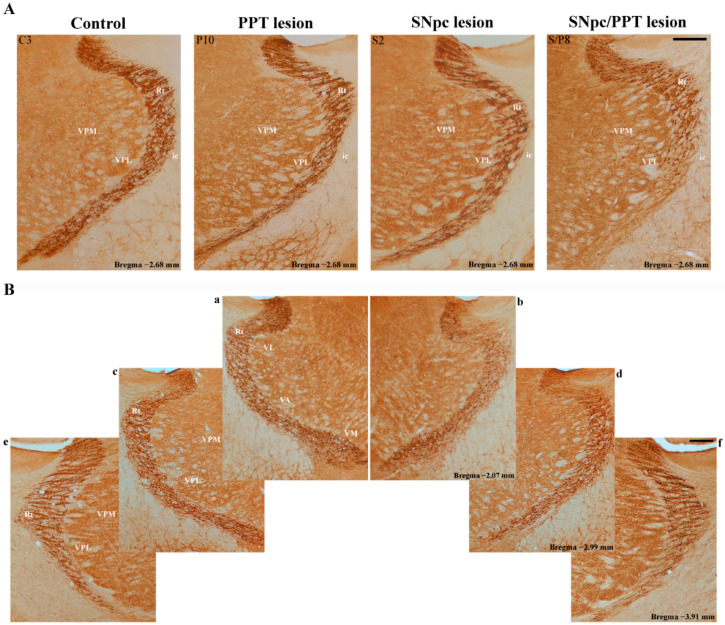
Individual examples of PV immunostaining alteration within RT in the distinct rat models of PD neuropathology. (**A**) The typical individual examples of PV immunoreactivity within RT in the control rat (Control, C3), rat with PD cholinopathy (PPT lesion, P10), hemiparkinsonian rat (SNpc lesion, S2), and hemiparkinsonian rat with PD cholinopathy (SNpc/PPT lesion, S/P8; see also Appendix A); (**B**) The antero-posterior dimension of PV immunoreactivity defect in the hemiparkinsonian rat with PD cholinopathy (SNpc/PPT lesion, S/P8) spreads from −2.07 to −3.91 mm posterior from bregma, ipsilaterally to the combined lesion (**b**,**d**,**f**), and vs. the contralateral brain side (**a**,**c**,**e**). Rt—reticular nucleus; VPM—ventral posteromedial thalamic nucleus; VPL—ventral posterolateral thalamic nucleus; ic—internal capsule. Scale bar is 200 μm. PV immunoreactivity was reduced within RT of the hemiparkinsonian rats with PD cholinopathy (SNpc/PPT lesion) ipsilaterally to the combined SNpc/PPT lesion (the right RT) vs. the contralateral RT (the left RT), and vs. the control rats, rats with PD cholinopathy and hemiparkinsonian rats.

**Figure 12 ijms-22-08922-f012:**
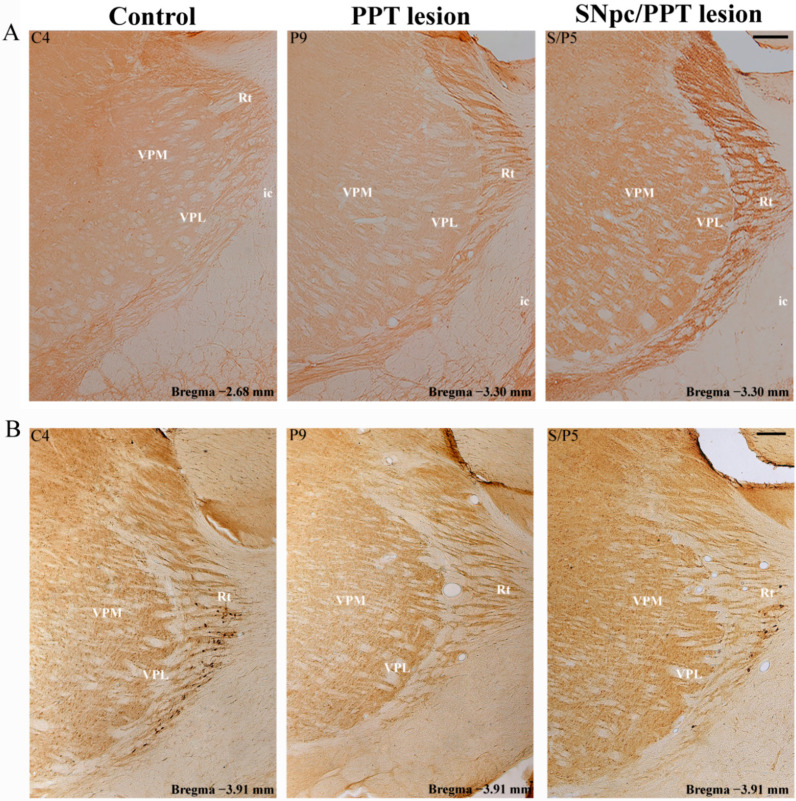
Individual examples of MAP2 and PSD-95 immunostaining alteration within RT in the distinct rat models of PD neuropathology. (**A**) The typical individual examples of MAP2 immunoreactivity within RT in the control rat (Control, C4), rat with PD cholinopathy (PPT lesion, P9), and hemiparkinsonian rat with PD cholinopathy (SNpc/PPT lesion, S/P5). Scale bar is 200 μm. (**B**) The typical individual examples of PSD-95 immunoreactivity within RT in the control rat (Control, C4), rat with PD cholinopathy (PPT lesion, P9), and hemiparkinsonian rat with PD cholinopathy (SNpc/PPT lesion, S/P5). Rt—reticular nucleus; VPM—ventral posteromedial thalamic nucleus; VPL—ventral posterolateral thalamic nucleus; ic— internal capsule. Scale bar is 200 μm. Whereas the partially increased MAP2 expression in RT of the rats with PD cholinopathy was followed by the suppression of PSD-95 expression, the enhanced MAP2 immunoreactivity in RT of the hemiparkinsonian rats with PD cholinopathy was followed by PSD-95 expression similar to control level.

**Table 1 ijms-22-08922-t001:** Sleep spindles (SS) dynamic during NREM and REM sleep in the distinct rat models of PD.

	**Control**							
		***n***	**nSS**	**Density/h** **(1/min)**	**NREM/REM dur/h (min)**	**SSdur/h (min)**	**SSdur** **(s)**	**SSf** **(Hz)**
**MCx**	***NREM***	5/8 (63%)	29	0.34 ± 0.17	25.70 ± 4.28	0.13 ± 0.04	1.34 ± 0.12	13.56± 0.21
	***REM***	7/8 (88%)	17	0.38 ± 0.11	7.00 ± 0.74	0.06 ± 0.02	1.39 ± 0.18	13.65 ± 0.28
**Hipp**	***NREM***	3/8 (38%)	12	\	\	\	\	\
	***REM***	3/8 (38%)	4	\	\	\	\	\
	**PPT lesion**							
		***n***	**nSS**	**Density/h** **(1/min)**	**NREM/REM dur/h (min)**	**SSdur/h (min)**	**SSdur** **(s)**	**SSf** **(Hz)**
**MCx**	***NREM***	8/8 (100%)	124	0.52 ± 0.10	31.02 ± 2.33	0.31 ± 0.05	1.18 ± 0.04	13.24 ± 0.09
	***REM***	8/8 (100%)	18	0.22 ± 0.07 *	10.69 ± 1.40	0.06 ± 0.02	1.52 ± 0.16 *	12.69 ± 0.16
**Hipp**	***NREM***	6/8 (75%)	31	**0.17 ± 0.08**	31.81 ± 2.91	0.10 ± 0.04	1.21 ± 0.06	**12.70 ± 0.10**
	***REM***	2/8 (25%)	6	\	\	\	\	\
	**SNpc lesion**							
		***n***	**nSS**	**Density/h** **(1/min)**	**NREM/REM dur/h (min)**	**SSdur/h (min)**	**SSdur** **(s)**	**SSf** **(Hz)**
**MCx**	***NREM***	5/7 (71%)	132	0.91 ± 0.21	28.30 ± 4.10	0.54 ± 0.16	1.23 ± 0.05	13.31 ± 0.08
	***REM***	5/7 (71%)	34	0.73 ± 0.21	9.00 ± 0.46	0.21 ± 0.06	1.88 ± 0.14 *	13.14 ± 0.11
**Hipp**	***NREM***	5/7 (71%)	36	0.24 ± 0.07	28.83 ± 3.73	0.14 ± 0.05	1.19 ± 0.08	**12.56 ± 0.11**
	***REM***	3/7 (43%)	9	\	\	\	\	\
	**SNpc/PPT lesion**							
		***n***	**nSS**	**Density/h** **(1/min)**	**NREM/REM dur/h (min)**	**SSdur/h (min)**	**SSdur** **(s)**	**SSf** **(Hz)**
**MCx**	***NREM***	8/8 (100%)	166	0.81 ± 0.19	25.88 ± 3.05	0.44 ± 0.16	1.28 ± 0.04	13.27 ± 0.07
	***REM***	5/8 (63%)	21	0.52 ± 0.16	8.67 ± 1.30	0.10 ± 0.02	1.47 ± 0.11	13.40 ± 0.22
**Hipp**	***NREM***	3/8 (38%)	19	\	\	\	\	\
	***REM***	5/8 (63%)	12	0.33 ± 0.06	7.57 ± 1.20	0.06 ± 0.01	1.51 ± 0.14	**12.51 ± 0.38**

*n*—number of rats; nSS—total number of extracted SS used for analysis of SS dynamic; Density/h—mean frequency of SS per hour of NREM/REM sleep (1/min); NREM/REM dur/h—mean NREM/REM duration (minutes); SSdur/h—total duration of SS extracted during 1 h of NREM/REM sleep (minutes); SSdur—mean SS duration (seconds); SSf—mean SS intrinsic frequency (Hz). Bold numbers indicate the statistically significant difference of the inter-structure (MCx vs. Hipp) mean values within the corresponding group at *p* ≤ 0.05; * indicates the statistically significant difference of the NREM/REM mean values at *p* ≤ 0.05. All comparisons were made using the Mann–Whitney U-test. Backslashes indicate the mean values that were not calculated and compared since the SSs were present in less than 50% of the rats.

**Table 2 ijms-22-08922-t002:** High voltage sleep spindles (HVS) dynamic during NREM and REM sleep in the distinct rat models of PD.

	**Control**							
		***n***	**nHVS**	**Density/h** **(1/min)**	**NREM/REM dur/h (min)**	**HVSdur/h (min)**	**HVSdur** **(s)**	**HVSf** **(Hz)**
**MCx**	***NREM***	1/8 (13%)	2	\	\	\	\	\
	***REM***	3/8 (38%)	14	\	\	\	\	\
**Hipp**	***NREM***	1/8 (13%)	3	\	\	\	\	\
	***REM***	2/8 (25%)	25	\	\	\	\	\
	**PPT lesion**							
		***n***	**nHVS**	**Density/h** **(1/min)**	**NREM/REM dur/h (min)**	**HVSdur/h (min)**	**HVSdur** **(s)**	**HVSf** **(Hz)**
**MCx**	***NREM***	7/8 (88%)	33	0.16 ± 0.05	31.17 ± 2.69	0.12 ± 0.04	1.57 ± 0.08	7.67 ± 0.19
	***REM***	8/8 (100%)	79	1.15 ± 0.40 *	10.69 ± 1.40	0.46 ± 0.09	2.79 ± 0.12 *	7.53 ± 0.09
**Hipp**	***NREM***	7/8 (88%)	16	0.07 ± 0.02	30.64 ± 2.66	0.07 ± 0.03	1.74 ± 0.11	**7.11 ± 0.17**
	***REM***	8/8 (100%)	104	1.20 ± 0.26 *	10.69 ± 1.40	0.56 ± 0.19	2.58 ± 0.09 *	**7.17 ± 0.07**
	**SNpc lesion**							
		***n***	**nHVS**	**Density/h** **(1/min)**	**NREM/REM dur/h (min)**	**HVSdur/h (min)**	**HVSdur** **(s)**	**HVSf** **(Hz)**
**MCx**	***NREM***	3/7 (43%)	9	\	\	\	\	\
	***REM***	5/7 (71%)	69	1.56 ± 0.20	8.97 ± 0.47	0.61 ± 0.09	2.66 ± 0.14	7.68 ± 0.10
**Hipp**	***NREM***	1/7 (14%)	1	\	\	\	\	\
	***REM***	5/7 (71%)	35	**0.78 ± 0.13**	8.97 ± 0.47	0.30 ± 0.04	2.54 ± 0.18	**7.36 ± 0.11**
	**SNpc/PPT lesion**							
		***n***	**nHVS**	**Density/h** **(1/min)**	**NREM/REM dur/h (min)**	**HVSdur/h (min)**	**HVSdur** **(s)**	**HVSf** **(Hz)**
**MCx**	***NREM***	7/8 (88%)	87	0.47 ± 0.12	26.12 ± 3.51	0.33 ± 0.12	1.60 ± 0.01	7.61 ± 0.01
	***REM***	8/8 (100%)	52	0.94 ± 0.21	7.94 ± 1.18	0.26 ± 0.06	2.39 ± 0.11 *	7.93 ± 0.09 *
**Hipp**	***NREM***	3/8 (38%)	6	\	\	\	\	\
	***REM***	8/8 (100%)	56	0.92 ± 0.26	7.94 ± 1.18	0.28 ± 0.09	2.36 ± 0.11	7.61 ± 0.12

*n*—number of rats; nHVS—total number of extracted HVS used for analysis of HVS dynamic; density/h—mean frequency of HVS per hour of NREM/REM sleep (1/min); NREM/REM dur/h—mean NREM/REM duration (minutes); HVSdur/h—total duration of HVS extracted during 1 h of NREM/REM sleep (minutes); HVSdur—mean HVS duration (seconds); HVSf—mean HVS intrinsic frequency (Hz). Bold numbers indicate the statistically significant difference of the inter-structure (MCx vs. Hipp) mean values within the corresponding group at *p* ≤ 0.05; * indicates the statistically significant difference of the NREM/REM mean values at *p* ≤ 0.05. All comparisons were made using the Mann–Whitney U test. Backslashes indicate the mean values that were not calculated and compared, since the HVSs were present in less than 50% of the rats.

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
