# Peer review of "Hippocampal and Reticulo-Thalamic Parvalbumin Interneurons and Synaptic Re-Organization during Sleep Disorders in the Rat Models of Parkinson’s Disease Neuropathology"

_ijms, 2021, doi:10.3390/ijms22168922_

Round 1

Reviewer 1 Report

Radovanovic et al. investigated a role of parvalbumin-positive GABAergic neurons in the hippocampus and reticulo-thalamic nucleus related to sleep disorders as a prodromal symptom in Parkinson’s disease. Although the present study revealed synaptic alterations involved in neuropathology of Parkinson’s disease using animal models. there are some points that should be revised as folloes:

Introduction

  1. Throughout this section, I have no idea what the authors mean by such as "distinct model rat" and “the rat model of PD cholinopathy”. Therefore, we cannot understand the previous reports that the authors repeatedly refer to. Readers will be confused if the authors do not mention in this section what kind of PD model this rat is. Explanations of background and pathology of the model, such as the one at the beginning of the "Results" section, should be given in the "Introduction".

  1. Lines 52-64. In this paragraph, the authors repeat the same assertion that various sleep disorders including RBD are observed as prodromal symptoms in PD. Brush up on your sentence structure.

  1. Lines 88-89. Even though the two previous paragraph (lines 52-64) focuses on sleep disturbances as the most frequent non-motor symptom in PD, why does this paragraph conclude that the contribution of GABAergic dysfunction to cognitive function should be clarified? In addition, the next paragraph moves on to sleep again and cholinergic nerves so the meaning of each paragraph in the whole is unclear.

Results

  1. Line 153. What is “PPT” ? Please clarify the definition.

  1. Figures 1-2. Why did the authors quantify PV-positive neurons in DG with three different bregma points, anterior, medium, and posterior? Is there any difference in the strength of neural connectivity with the lesioned areas? If so, is there any correlation between the strength and the degree of change in model animals?

  1. Figures 7-12. The annotations in the panels appear to be somewhat small and difficult to read. It would be better to make them larger, or if possible, put a larger brain atlas of the relevant area and add annotations to it, not image itself.

  1. IHC bright-field staining. While PV-positive cells are quantified, MAP2 and PSD95 levels are not and the quality of the representative images in the present study are not enough to explain the alterations the authors stated. I think the authors should show images with higher magnification, or shift to fluorescently double-labeling of PSD-95 with MAP2 or presynaptic markers so that the immunoreactivity is more visible.

Discussion

  1. Lines 475-562. These paragraphs merely restate the results from the present and previous studies. Discuss the possible implications of these results, or else they should not be stated as already mentioned in “Results”.

  1. Lines 580-582. Even though it has been suggested that sleep disturbance as a prodromal symptom in PD accelerates the neurodegenerative process, the present study does not provide such evidence.

  1. Lines 589-594 and 600-603. It is better to add citations to support these statements.

  1. Lines 589-612. What the authors are trying to argue in these paragraphs is that parvalbumin-positive cells, as calcium-buffering GABAergic interneurons, are important for synaptic function? If so, in the present study why did the authors focus only on parvalbumin and not on calbindin or calretinin, which are the same calcium-binding GABAergic interneurons?

  1. Lines 604-612. In this and other paragraphs the authors suggest that excitation/inhibition imbalance induced by reduced PV-positive GABAergic interneurons is related to behavioral and cellular impairments; however, reduction in PV expression and that in PV-positive neuron itself have the opposite meanings regarding the electrophysiology. Which one PV reduction represents in the present study?

  1. Lines 631-648. The authors seem to consider the reticulo-thalamic nucleus as an important region involved in sleep disorders. I wonder there is any relationship between the neuropathology in the region and symptomatic progression in PD since expression and progression of various symptoms including prodromal ones in PD are reported to be well correlated with neuropathology (such as α-synuclein burden) in CNS. If so, the reticulo-thalamic nucleus could be a notable region related to the onset of PD also clinically.

  1. Lines 649-650. The authors repeatedly suggest that a reduction in PV consequently enhances MAP2 expression and synaptic re-organization; however, the present study has little evidence that PV reduction but not MAP2 enhancement precedes with the other two alterations. Is there any evidence from the results or previous reports?

  1. It remains unclear why the lesions cause changes in PV, MAP2, and PSD95 levels and the differences between two models. Is there any evident data or previous report with mechanisms at cellular levels?

Author Response

Radovanovic et al. investigated a role of parvalbumin-positive GABAergic neurons in the hippocampus and reticulo-thalamic nucleus related to sleep disorders as a prodromal symptom in Parkinson’s disease. Although the present study revealed synaptic alterations involved in neuropathology of Parkinson’s disease using animal models. There are some points that should be revised as follows:

 Introduction

  1. Throughout this section, I have no idea what the authors mean by such as "distinct model rat" and “the rat model of PD cholinopathy”. Therefore, we cannot understand the previous reports that the authors repeatedly refer to. Readers will be confused if the authors do not mention in this section what kind of PD model this rat is. Explanations of background and pathology of the model, such as the one at the beginning of the "Results" section, should be given in the "Introduction".

We now added an explanation of our distinct rat models of PD, used as the experimental groups in this study, in Introduction p.2, the last paragraph; p.3, the third paragraph, and p.4, the first paragraph; beside the explanations in Results, Section 2.1, p.4, the second paragraph, and in Materials and Methods, Section 4.1.

  1. Lines 52-64. In this paragraph, the authors repeat the same assertion that various sleep disorders including RBD are observed as prodromal symptoms in PD. Brush up on your sentence structure.

 We corrected this paragraph in Introduction. Please see p.2, the second paragraph.

  1. Lines 88-89. Even though the two previous paragraph (lines 52-64) focuses on sleep disturbances as the most frequent non-motor symptom in PD, why does this paragraph conclude that the contribution of GABAergic dysfunction to cognitive function should be clarified? In addition, the next paragraph moves on to sleep again and cholinergic nerves so the meaning of each paragraph in the whole is unclear.

We revised Introduction and corrected this sentence, at p.2, the end of the fourth paragraph.

 Results

  1. Line 153. What is “PPT” ? Please clarify the definition.

We now clarified the abbreviation “PPT” in Introduction. Please see p.2, the second sentence of the last paragraph, and first line on p.3.

  1. Figures 1-2. Why did the authors quantify PV-positive neurons in DG with three different bregma points, anterior, medium, and posterior? Is there any difference in the strength of neural connectivity with the lesioned areas? If so, is there any correlation between the strength and the degree of change in model animals?

We quantified PV+ interneurons in DG throughout the overall antero-posterior dimension of the hippocampus by using the ImageJ 1.46 and .tiff images. We used three tissue samples per each rat of each experimental group and grouped them in three defined stereotaxic ranges.

All details of PV+ interneurons quantification are explained in Materials and Methods, Section 4.5.3.

Since we used 40 µm sections for immunostaining and we counted the number of PV+ interneurons on .tiff figures, our quantification of the PV+ interneurons in DG, even throughout the overall antero-poterior dimension of hippocampus, is more estimated PV+ interneurons number than the real absolute number of PV+ interneurons within the overall depth of tissue sections.

The topographical differences of the impact of PPT, SNpc, or SNpc/PPT lesions (deafferentations) are possible either at hippocampal or thalamic level.

We presented the results of our study as the group results on Fig. 1, and their corresponding individual examples per each experimental group and per each stereotaxic range on Fig. 2.

For any detailed afferent-efferent correlation the detailed tracing studies would be necessary.

  1. Figures 7-12. The annotations in the panels appear to be somewhat small and difficult to read. It would be better to make them larger, or if possible, put a larger brain atlas of the relevant area and add annotations to it, not image itself.

We agree. In the template file of our submitted manuscript the size of our Figures were reduced from the originals.

Now, in this revised version of the manuscript we increased the size of all figures in the submitted template file and hopefully we improved them.

But, please see also all initially submitted original Figs. of this manuscript that we have submitted as .tiff files separately from the manuscript template file.

  1. IHC bright-field staining. While PV-positive cells are quantified, MAP2 and PSD95 levels are not and the quality of the representative images in the present study are not enough to explain the alterations the authors stated. I think the authors should show images with higher magnification, or shift to fluorescently double-labeling of PSD-95 with MAP2 or presynaptic markers so that the immunoreactivity is more visible.

Now, in this revised manuscript we increased the size of all figures in the submitted manuscript template file and hopefully we improved figures visibility.

We have also submitted new Fig. 9 and Fig. 10 with higher magnified representative images - scale bar is now 25 µm instead of 50 µm.

But, please see also all initially submitted original Figs. of this manuscript that we submitted as .tiff files separately from the manuscript template file.

Discussion

  1. Lines 475-562. These paragraphs merely restate the results from the present and previous studies. Discuss the possible implications of these results, or else they should not be stated as already mentioned in “Results”.

 We corrected this and excluded the fourth and fifth paragraph of the Discussion.

  1. Lines 580-582. Even though it has been suggested that sleep disturbance as a prodromal symptom in PD accelerates the neurodegenerative process, the present study does not provide such evidence.

We agree. Our study does not provide such evidence. We only discussed the other studies related to the same issue that are in accordance to our present results.

  1. Lines 589-594 and 600-603. It is better to add citations to support these statements.

We corrected this paragraph and added the citations.

  1. Lines 589-612. What the authors are trying to argue in these paragraphs is that parvalbumin-positive cells, as calcium-buffering GABAergic interneurons, are important for synaptic function? If so, in the present study why did the authors focus only on parvalbumin and not on calbindin or calretinin, which are the same calcium-binding GABAergic interneurons?

So far, in this study we did first the parvalbumin immunostaining at the hippocampal and reticulo-thalamic level, but in our future studies it is possible to do the other calcium-binding GABA interneurons.

  1. Lines 604-612. In this and other paragraphs the authors suggest that excitation/inhibition imbalance induced by reduced PV-positive GABAergic interneurons is related to behavioral and cellular impairments; however, reduction in PV expression and that in PV-positive neuron itself have the opposite meanings regarding the electrophysiology. Which one PV reduction represents in the present study?

We discussed this issue in Discussion, p. 22, the last paragraph; p.23, the first paragraph; p. 24, the third and last paragraph; p.25, the first and second paragraph

  1. Lines 631-648. The authors seem to consider the reticulo-thalamic nucleus as an important region involved in sleep disorders. I wonder there is any relationship between the neuropathology in the region and symptomatic progression in PD since expression and progression of various symptoms including prodromal ones in PD are reported to be well correlated with neuropathology (such as α-synuclein burden) in CNS. If so, the reticulo-thalamic nucleus could be a notable region related to the onset of PD also clinically.

Yes, our present study indicates the RT, an important brain structure for sleep spindle generation and state control, also as an important brain structure for the onset of PD.

  1. Lines 649-650. The authors repeatedly suggest that a reduction in PV consequently enhances MAP2 expression and synaptic re-organization; however, the present study has little evidence that PV reduction but not MAP2 enhancement precedes with the other two alterations. Is there any evidence from the results or previous reports?

We do not have that evidence. To our knowledge there is no evidence in previous reports.

  1. It remains unclear why the lesions cause changes in PV, MAP2, and PSD95 levels and the differences between two models. Is there any evident data or previous report with mechanisms at cellular levels?

We agree. Further studies will hopefully resolve all these issue, but so far we concluded, based on our present results:

”Our study demonstrates for the first time an important regulatory role of the hippocampal and RT GABAergic PV+ interneurons on the synaptic protein dynamic alterations in distinct rat models of PD neuropathology which are reflected prodromally, distinctly and long-lasting at functional level: from distinct local sleep disorders through distinct alteration of sleep–related EEG oscillations to distinct alteration of the sleep spindles dynamics. Our results in the rat models of PD neuropathology indicate that by augmenting the GABAergic signaling via PV+ interneuron modulation can be effective in improving or ameliorating prodromal sleep disorders and memory deficits in PD”.

Reviewer 2 Report

The Authors, Drs Radovanovic et al., submitted an article in which they investigated the alterations of hippocampal and reticulo-thalamic GABAergic parvalbumin interneurons and their synaptic re-organizations underlying the prodromal local sleep disorders in the distinct rat models of Parkinson’s Disease.

The manuscript is scientifically interesting. Paragraphs are well organized, and the purpose of the article is clear.

However, the title is too long and the English language needs revisions.

Author Response

The Authors, Drs Radovanovic et al., submitted an article in which they investigated the alterations of hippocampal and reticulo-thalamic GABAergic parvalbumin interneurons and their synaptic re-organizations underlying the prodromal local sleep disorders in the distinct rat models of Parkinson’s Disease.

The manuscript is scientifically interesting. Paragraphs are well organized, and the purpose of the article is clear.

However, the title is too long and the English language needs revisions.

Thank you so much for your comments.

We revised the title and English language of our manuscript.

All our corrections of manuscript text are in track changes of the manuscript template file submitted to your journal.

Reviewer 3 Report

In the current manuscript, Radovanovic et al investigated alterations of hippocampal and reticulo-thalamic (RT) PV+ interneurons and their synaptic re-organization underlying prodromal sleep disorders in diverse rat models of Parkinson’s disease (PD).

The authors have employed histological, physiological and behavioral tools to address their research question. A lot of data was collected for this study, going to show that a substantial volume of work was performed.  In order to get this work published, the authors would need to address some questions that have come up during the review. I suggest major revisions for now.

Major issues:

  1. In Figure 1, I am not convinced by increased PV cell counts in rats with PPT lesion. This is because @-4.7 to 6.2 mm from the bregma, the increase in PV cell numbers is not evident. From the images shown in Figure 2, it is also not clear if there was an increase in PV IN numbers @-3.1 to 4.5 mm in PPT lesioned rats. I am also surprised looking at the PV immunostaining, especially the number of PV INs. Number of PV INs seems to be much lower.
  2. Have the authors looked into the morphology of PV INs? It would be interesting if the authors could show if there are morphological deficits in PV INs in PPT vs SNc/PPT lesioned animals.
  3. I am not convinced by the locomotor activity quantified by the authors as a behavioral proxy of their histological data. If the authors were interested in spatial memory, it would have been better if they performed specific spatial memory dependent tasks, for example T-maze/Morris water maze. Data presented in Figure 6 should therefore be complemented with another employing a more relevant behavioral task.
  4. PSD-95 immunostaining is not convincing, at least in the FOVs that have been shown to us in Figure 9. I would like to see some zoomed section of an image to apprehend the efficacy of PSD-95 staining.

Minor issues:

  1. The manuscript needs professional editing help. There are several grammatical errors (for example, usage of ‘the’ in sentences) throughout the manuscript that needs correction.
  2. If abbreviations are used, the authors should make clear to the readers what those abbreviated words are. One instance, I did not find what DG was in the entire manuscript.
  3. Number of tissues used/animal should be mentioned in the figure legends along with the statistical test employed. This helps a reader understand the data well.

Author Response

In the current manuscript, Radovanovic et al investigated alterations of hippocampal and reticulo-thalamic (RT) PV+ interneurons and their synaptic re-organization underlying prodromal sleep disorders in diverse rat models of Parkinson’s disease (PD).

The authors have employed histological, physiological and behavioral tools to address their research question. A lot of data was collected for this study, going to show that a substantial volume of work was performed.  In order to get this work published, the authors would need to address some questions that have come up during the review. I suggest major revisions for now.

Major issues:

  1. In Figure 1, I am not convinced by increased PV cell counts in rats with PPT lesion. This is because @-4.7 to 6.2 mm from the bregma, the increase in PV cell numbers is not evident. From the images shown in Figure 2, it is also not clear if there was an increase in PV IN numbers @-3.1 to 4.5 mm in PPT lesioned rats. I am also surprised looking at the PV immunostaining, especially the number of PV INs. Number of PV INs seems to be much lower.

We quantified PV+ interneurons in DG throughout the overall antero-posterior dimension of the hippocampus by using the ImageJ 1.46 and .tiff figures. We used three tissue samples per each rat of each experimental group, grouped in three defined stereotaxic ranges.

All details of PV+ interneurons quantification are explained in Materials and Methods, Section 4.5.3.

Since we used 40 µm sections for immunostaining and we counted the number of PV+ interneurons on .tiff figures, our quantification of the PV+ interneurons in DG even throughout the overall antero-poterior dimension of hippocampus is more estimated number than the real absolute number of PV+ interneurons within the overall depth of tissue sections.

The topographical differences of the impact of the PPT, SNpc, or SNpc/PPT lesions (deafferentations) are possible either at hippocampal or thalamic level.

We presented the results of our study as the group results on Fig. 1, and their corresponding individual examples per each experimental group and per each stereotaxic range on Fig. 2.

For any detailed topographical afferent-efferent correlations the detailed tracing studies will be necessary.

  1. Have the authors looked into the morphology of PV INs? It would be interesting if the authors could show if there are morphological deficits in PV INs in PPT vs SNc/PPT lesioned animals.

No, we did not follow the morphological changes of the PV+ interneurons in this study, but we agree that this is also a very important and interesting issue for our future studies.

  1. I am not convinced by the locomotor activity quantified by the authors as a behavioral proxy of their histological data. If the authors were interested in spatial memory, it would have been better if they performed specific spatial memory dependent tasks, for example T-maze/Morris water maze. Data presented in Figure 6 should therefore be complemented with another employing a more relevant behavioral task.

We agree. We discussed this issue, or the limitation of our present study, at p. 22, the third paragraph, but future studies are needed to resolve this issue.

  1. PSD-95 immunostaining is not convincing, at least in the FOVs that have been shown to us in Figure 9. I would like to see some zoomed section of an image to apprehend the efficacy of PSD-95 staining.

We agree. Now, in this revised manuscript, we increased the size of all figures in the submitted manuscript template file, and hopefully we improved their visibility.

But, please see also all initially submitted original Figs. of this manuscript that we submitted as .tiff files separately from the template manuscript file.

We provide you the new Figs. 9, 10 with higher magnified images - scale bars are now 25 µm instead of 50 µm.

Minor issues:

  1. The manuscript needs professional editing help. There are several grammatical errors (for example, usage of ‘the’ in sentences) throughout the manuscript that needs correction.

We corrected them, particularly ‘the’ throughout the manuscript text.

  1. If abbreviations are used, the authors should make clear to the readers what those abbreviated words are. One instance, I did not find what DG was in the entire manuscript.

We did it.

  1. Number of tissues used/animal should be mentioned in the figure legends along with the statistical test employed. This helps a reader understand the data well.

We mentioned all the missing numbers and statistical tests.           

Round 2

Reviewer 1 Report

Radovanovic et al. have responded to my comments and submitted some points that need to be revised. Although I now generally agree with the present study, it is insufficiently revised in several points. I would like you to understand the meaning of the comments again and respond to them carefully.

  1. Previous comment: Figures 1-2. Why did the authors quantify PV-positive neurons in DG with three different bregma points, anterior, medium, and posterior? Is there any difference in the strength of neural connectivity with the lesioned areas? If so, is there any correlation between the strength and the degree of change in model animals?

Authors’ response: We quantified PV+ interneurons in DG throughout the overall antero-posterior dimension of the hippocampus by using the ImageJ 1.46 and .tiff images. We used three tissue samples per each rat of each experimental group and grouped them in three defined stereotaxic ranges.

All details of PV+ interneurons quantification are explained in Materials and Methods, Section 4.5.3.

Since we used 40 µm sections for immunostaining and we counted the number of PV+ interneurons on .tiff figures, our quantification of the PV+ interneurons in DG, even throughout the overall antero-poterior dimension of hippocampus, is more estimated PV+ interneurons number than the real absolute number of PV+ interneurons within the overall depth of tissue sections.

The topographical differences of the impact of PPT, SNpc, or SNpc/PPT lesions (deafferentations) are possible either at hippocampal or thalamic level.

We presented the results of our study as the group results on Fig. 1, and their corresponding individual examples per each experimental group and per each stereotaxic range on Fig. 2.

For any detailed afferent-efferent correlation the detailed tracing studies would be necessary.

Additional comment: My previous question was not how did you quantify PV-positive cells, but why did you quantify them in three parts according to bregma? Although the authors responded that further analysis will be needed on the neural connections between the PV-positive cells in the DG and the lesioned areas, but at the time of this quantification, what was the purpose of the 3-part division? More specifically, Figs. 3 and 4 evaluate the functional coupling between changes in PV-positive cells in DG and lesioned areas (Fig. 3) or wake episode duration (Fig. 4); however, the authors did not explain why these figures are shown with the overall PV-positive cells rather than the divided ones with the bregma.

  1. Previous comment: Lines 649-650. The authors repeatedly suggest that a reduction in PV consequently enhances MAP2 expression and synaptic re-organization; however, the present study has little evidence that PV reduction but not MAP2 enhancement precedes with the other two alterations. Is there any evidence from the results or previous reports?

Authors’ response: We do not have that evidence. To our knowledge there is no evidence in previous reports.

Additional comment: If there is no such evidence in the present study or in previous reports, the authors should not make assertions such as "synaptic re-organization following changes in PV-positive cells", because it is unclear which factor precedes the other, and whether there is a clear link between the two.

Author Response

Radovanovic et al. have responded to my comments and submitted some points that need to be revised. Although I now generally agree with the present study, it is insufficiently revised in several points. I would like you to understand the meaning of the comments again and respond to them carefully.

  1. Previous comment: Figures 1-2. Why did the authors quantify PV-positive neurons in DG with three different bregma points, anterior, medium, and posterior? Is there any difference in the strength of neural connectivity with the lesioned areas? If so, is there any correlation between the strength and the degree of change in model animals?

Authors’ response: 

We quantified PV+ interneurons in DG throughout the overall antero-posterior dimension of the hippocampus by using the ImageJ 1.46 and .tiff images. We used three tissue samples per each rat of each experimental group and grouped them in three defined stereotaxic ranges.

 All details of PV+ interneurons quantification are explained in Materials and Methods, Section 4.5.3.

 Since we used 40 µm sections for immunostaining and we counted the number of PV+ interneurons on .tiff figures, our quantification of the PV+ interneurons in DG, even throughout the overall antero-poterior dimension of hippocampus, is more estimated PV+ interneurons number than the real absolute number of PV+ interneurons within the overall depth of tissue sections.

The topographical differences of the impact of PPT, SNpc, or SNpc/PPT lesions (deafferentations) are possible either at hippocampal or thalamic level.

 We presented the results of our study as the group results on Fig. 1, and their corresponding individual examples per each experimental group and per each stereotaxic range on Fig. 2.

 For any detailed afferent-efferent correlation the detailed tracing studies would be necessary.

Additional comment: My previous question was not how did you quantify PV-positive cells, but why did you quantify them in three parts according to bregma? Although the authors responded that further analysis will be needed on the neural connections between the PV-positive cells in the DG and the lesioned areas, but at the time of this quantification, what was the purpose of the 3-part division? More specifically, Figs. 3 and 4 evaluate the functional coupling between changes in PV-positive cells in DG and lesioned areas (Fig. 3) or wake episode duration (Fig. 4); however, the authors did not explain why these figures are shown with the overall PV-positive cells rather than the divided ones with the bregma.

Thanks for your useful comments that enable us to improve our manuscript.

The purpose of counting the PV+ interneurons in three stereotaxic ranges of hippocampal DG was to estimate the PV+ interneurons in the overall antero-posterior dimension of hippocampal DG and thus be able to track the volume/dimension of possible alterations.

We have also estimated in the same way the cholinergic neuronal loss of PPT and the dopaminergic neuronal loss of SNpc in their three stereotaxic ranges within their overall antero-posterior dimensions.

For the accurate topographical analysis of the PPT or SNpc impacts on the PV+ interneurons expression and their impact on the synaptic re-organization over time and vice versa within the hippocampus or remotely in the RT needs more experimental work that we will plan for our future studies.  Since in this study we have a relatively small number of data for the topographical correlations, for the Figs. 3,4 we pulled our data and used all the % of the Ach or DA deficits (from all defined stereotaxic ranges), expressed with respect to control values at each corresponding stereotaxic range as 100 %, as well as the PV+ interneurons numbers from all defined stereotaxic ranges.

  1. Previous comment: Lines 649-650. The authors repeatedly suggest that a reduction in PV consequently enhances MAP2 expression and synaptic re-organization; however, the present study has little evidence that PV reduction but not MAP2 enhancement precedes with the other two alterations. Is there any evidence from the results or previous reports?

 Authors’ response: We do not have that evidence. To our knowledge there is no evidence in previous reports.

 Additional comment: If there is no such evidence in the present study or in previous reports, the authors should not make assertions such as "synaptic re-organization following changes in PV-positive cells", because it is unclear which factor precedes the other, and whether there is a clear link between the two.

Thanks for this comment. We agree and we corrected the term „following” into “and” in all subtitles of Results (Results 2.5; Results 2.5.1; Results 2.5.2; Results 2.7; Results 2.7.1; Results 2.7.2). In addition, the term „following” was also corrected throughout the manuscript text.

All the new corrections made during minor revision are highlighted.

Reviewer 3 Report

The authors have answered my queries. Although I am still not completely convinced by the PSD-95 staining, I am happy with the explanations provided for all other questions that I had raised. 

I would like to accept the manuscript, albeit with a word of caution. Please select your behavioral strategy/task more thoughtfully.

Author Response

The authors have answered my queries. Although I am still not completely convinced by the PSD-95 staining, I am happy with the explanations provided for all other questions that I had raised. 

I would like to accept the manuscript, albeit with a word of caution. Please select your behavioral strategy/task more thoughtfully.

Thanks a lot for all your comments that helped us to improve our manuscript.